# Little millet genome reveals evolutionary insights into tetraploid structure and genetic basis of micronutrient density

Krishna Kishore Gali [1], Kevin C. Koh [2], Tara Chellapilla Satyavathi [3], Ganapathy Kuyyamudi Nanaiah [3], K. B. Palanna [4], Morgan W. Kirzinger[5], Sandeep Nanjundappa [3], Sampath Perumal[2], Deekshitha Bomireddy[3], H. B. Mahesh [6], Harshal Eknath Patil[7], Raju Chaudhary [2], Loveleen Kaur Dhillon[1], Venkat Bandi[2], V. B. Reddy Lachagari[8], Surya Teja Veeramachaneni[3], Renuka Malipatil[3], Peng Gao [9], Shankar Pahari [9], Andrew G. Sharpe [2], Thomas D. Warkentin[1], Raju Soolanayakanahally [9] ✉, M. K. Prasannakumar[10] ✉, Nepolean Thirunavukkarasu[3] ✉ & Sateesh Kagale [5] ✉

Little millet is a hardy and nutrient-rich cereal which improves food and nutritional security in marginal environments. Despite its importance, genomic resources for this orphan crop have been limited. Here, we report a high quality, chromosome-scale genome assembly of little millet comprising 18 chromosomes and 59,045 genes. Eleven chromosomes are assembled from telomere to telomere, revealing an 850 Mb tetraploid genome that closely resembles broomcorn millet. Comparative analyses indicate early stages of diploidization, characterized by gene loss and subgenome-specific expression biases that vary across genes and tissues. Resequencing of 300 accessions uncovers extensive genetic diversity, including single-nucleotide polymorphism and structural variants. Genome-wide association studies identify genetic loci linked to grain micronutrient traits, including several associated with high iron content. These genomic and phenotypic resources provide a foundation for molecular breeding and marker-assisted selection, enabling the improvement of little millet as a climate-resilient crop to support global food and nutritional security.

Millets, often referred to as "wonder cereals", are resilient orphan crops that thrive in harsh and challenging environments. They are categorized into major millets, such as sorghum and pearl millet, and minor millets, which include foxtail millet (*Setaria italica* L.),

barnyard millet (*Echinochloa esculenta* L.), little millet (*Panicum sumatrense* Roth ex Roem. & Schult.), kodo millet (*Paspalum scrobiculatum* L.), broomcorn millet (*Panicum miliaceum* L.), teff (*Eragrostis tef* [Zucc.] Trotter), fonio millet (*Digitaria exilis* L.), Job's tears

[1]Crop Development Centre, Department of Plant Sciences, University of Saskatchewan, Saskatoon, SK, Canada. [2]Global Institute for Food Security, University of Saskatchewan, Saskatoon, SK, Canada. [3]Global Center of Excellence on Millets (Shree Anna), ICAR-Indian Institute of Millets Research, Hyderabad, India. [4]ICAR-AICRP on Small Millets, University of Agricultural Sciences, Bengaluru, Karnataka, India. [5]Aquatic and Crop Resource Development, National Research Council Canada, Saskatoon, SK, Canada. [6]Department of Genetics and Plant Breeding, University of Agricultural Sciences, Bengaluru, India. [7]Navsari Agricultural University, Waghai, Gujarat, India. [8]ATGC Biotech Pvt Ltd., Genome Valley, Hyderabad, Telangana, India. [9]Saskatoon Research and Development Centre, Agriculture and Agri-Food Canada, Saskatoon, SK, Canada. [10]Department of Plant Pathology, University of Agricultural Sciences, Bengaluru, Karnataka, India. ✉e-mail: raju.soolanayakanahally@agr.gc.ca; babu_prasanna@rediffmail.com; tnepolean@gmail.com; Sateesh.Kagale@nrc-cnrc.gc.ca

(*Coix lacryma-jobi* L.), brown top millet (*Urochloa ramosa* L.), and guinea millet (*Brachiaria deflexa* L.)[1]. Millets provide substantial health benefits[2] and play a vital role in enhancing global food and nutritional security, particularly in developing regions. These nutrient-dense grains are rich sources of micronutrients, complex vitamins, and essential amino acids, which are often deficient in conventional diets, thereby boosting their nutritional value. Millets are also rich in energy supplying carbohydrates and dietary fiber. The growing preference for multigrain meals has reignited interest in these ancient crops, which once formed the cornerstone of traditional agriculture.

Little millet, a member of the *Poaceae* family (Fig. 1A–C), is among the underutilized minor millet species owing to poor support for production, research, and development[1,3]. Native to India, it is cultivated as an annual cereal crop in eastern Indonesia, Myanmar, and the semi-arid and hilly regions of India and Pakistan[4,5]. Little millet's growing popularity stems from its nutritional value, and adaptability to

marginal soils with minimal agricultural inputs. Renowned for its resilience, little millet demonstrates high tolerance to drought, salinity, and waterlogging, coupled with low susceptibility to pests and diseases. These traits make it an invaluable resource for smallholder farmers in resource-constrained regions[6,7]. Beyond its climate-resilience, little millet is especially rich in iron and dietary fiber[8] which makes it a valuable ingredient in multigrain and gluten-free cereal products. Little millet is also an excellent source of protein and other essential micronutrients, including calcium, zinc, and iodine[9] (Fig. 1D). Little millet holds great promise for promoting sustainable agriculture and healthier diets, serving as both a substitute and supplementary crop to enhance dietary diversity and bolster food and nutritional security. However, unlocking its full potential requires substantial improvements to grain productivity and quality. Advanced genetic and genomic tools are indispensable for understanding the genetic basis of key agronomic traits, including micronutrient concentration, and for deciphering the structure, organization, and expression patterns of

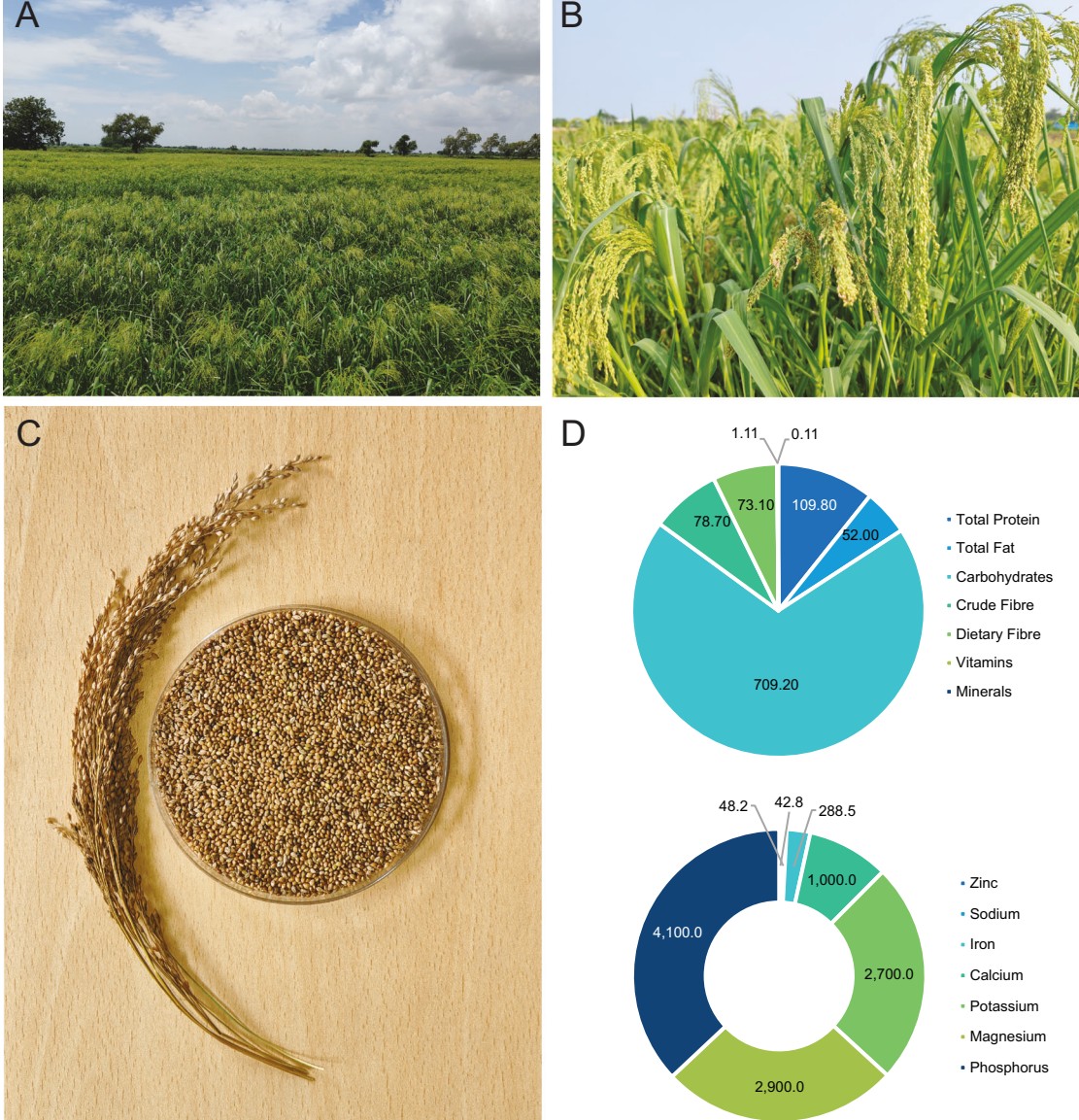

**Fig. 1 | Overview of little millet crop and its grain nutritional composition.**
**A** Field grown little millet. **B** Developing panicles. **C** Panicle and grain at harvest stage. **D** The nutritional profile (g/Kg seed) of the JK-8 variety. The JK-8 grains are primarily composed of carbohydrates (70.9 %), followed by protein (10.9 %), dietary fiber (7.3 %), and fat (5.2 %). The vitamin concentration (A, B, D, E, and K) was found to be 1.05 g/kg (0.1%). Among micronutrients, phosphorus (P) was the most abundant at 4100 ppm, followed by magnesium (Mg) at 2900 ppm, potassium (K) at 2700 ppm, and calcium (Ca) at 1000 ppm. Iron (Fe) was present at 288 ppm, while zinc (Zn) was recorded at 43 ppm. Source data are provided as a Source Data file.

underlying genes. Despite its importance, genomic information for little millet is limited, with fundamental aspects such as genome size and ploidy level[10] still remaining ambiguous.

A complete genome sequence of little millet would provide a foundation for identifying beneficial alleles for crop improvement. Recently, the genome of little millet cv. OLM-20, with 279 scaffolds, an N50 of 7.8 Mb was developed through long-read sequencing and Hi-C scaffolding[11]. However, the final scaffolded genome assembly of 566 Mb is significantly shorter than the 2.06 Gb of total bases assembled from PacBio reads, raising concerns about the assembly's completeness and accuracy. Furthermore, fundamental questions about the ploidy level of little millet remain unresolved. Recent studies have begun to elucidate the genetic architecture of little millet including transcriptome analyses that uncovered genes associated with metabolism, micronutrient enrichment, environmental stress regulation[9], as well as salt and drought tolerance[10,12]. While these efforts have shed light on stress tolerance mechanisms, the genetic regulation of grain mineral nutrition in little millet remains largely unexplored.

In this study, we present a high-quality genome assembly of little millet, and investigate the genetic basis of grain micronutrient concentration. Using PacBio Hi-Fi, Oxford Nanopore long-read sequencing, and Hi-C sequencing, we assemble a nearly complete genome of little millet cv. JK-8, consisting of 18 chromosomes. Eleven chromosomes are assembled from telomere-to-telomere into single contigs, while the remaining chromosomes are split into 2–5 contigs each. Comparative genomic analysis with other millet species reveals a tetraploid structure in little millet, and a shared evolutionary trajectory with broomcorn millet. Using a panel of 300 little millet germplasm accessions phenotyped at two locations, significant marker-trait associations for grain iron content, content of other minerals, and key agronomic traits are identified. These findings provide critical genomic resources and insights to enhance little millet's genetic improvement, paving the way for its development as a resilient and nutritionally enriched staple crop.

## Results

### Genome sequencing and assembly

We sequenced the genome of little millet cv. JK-8, a pure line generated through single seed descent, using a hybrid approach combining PacBio Hi-Fi and Oxford Nanopore Technologies (ONT) data. The PacBio Hi-Fi sequencing yielded 70.1 Gb of long reads, providing ~82-fold coverage of the genome (Supplementary Table 1), estimated to be approximately 832 Mb based on k-mer statistics (Supplementary Fig. 1). The initial assembly using Hifiasm[13] resulted in 2019 scaffolds, covering a total length of 883.6 Mb, with N50 and N90 contig sizes of 26.8 Mb and 6.05 Mb, respectively (Supplementary Table 2). To improve assembly contiguity, we generated an additional 144 Gb of ONT data. Integration of ONT data with PacBio Hi-Fi reads using Hifiasm yielded an enhanced assembly of 351 contigs spanning 850.6 Mb, with improved N50 and N90 contig sizes of 29.4 Mb and 13 Mb, respectively (Supplementary Table 2). Next, we applied Hi-C proximity ligation scaffolding (Supplementary Fig. 2), which organized the 37 longest contigs, representing 97.7% of the total assembly bases (830.6 Mb), into 18 pseudomolecules with 19 gaps (Version 1 assembly). The remaining 314 smaller contigs, totaling 19.9 Mb, were left unanchored (Supplementary Table 2). Subsequently, we used TGS-GapCloser with the available ONT and PacBio reads, which closed 8 of the 19 gaps, specifically in chromosomes 4A, 4B, 5B, 6A, 7A and 9B. Many of the filled gaps were relatively small, ranging from ~200 bp to 1 Kb. The gap-filling resolved chromosomes 4B, 6A, 7A and 9B into single-contig assemblies. The final assembly (version 2) contains only 11 gaps, with eleven chromosomes assembled as single telomere-to-telomere (T2T) contigs, five chromosomes formed from two to three contigs each, and two chromosomes formed from three and five contigs (Table 1). Putative centromere locations were identified on all

**Table 1 | Summary of little millet genome assembly**

| Chromosome | Length (bp) | No. of anchored contigs | Contig N50 (bp) |
|---|---|---|---|
| Chr1A | 43,243,116 | 2 | 29,128,507 |
| Chr1B | 43,137,277 | 3 | 22,009,687 |
| Chr2A[a] | 53,162,992 | 1 | 53,162,992 |
| Chr2B[a] | 53,115,015 | 1 | 53,115,015 |
| Chr3A[a] | 61,397,489 | 1 | 61,397,489 |
| Chr3B | 50,789,506 | 2 | 32,076,622 |
| Chr4A | 36,975,915 | 2 | 31,037,868 |
| Chr4B[a] | 35,829,819 | 1 | 35,829,819 |
| Chr5A[a] | 47,000,000 | 1 | 47,000,000 |
| Chr5B | 50,323,433 | 2 | 26,743,900 |
| Chr6A[a] | 41,209,881 | 1 | 41,209,881 |
| Chr6B[a] | 37,846,408 | 1 | 37,846,408 |
| Chr7A[a] | 39,612,390 | 1 | 39,612,390 |
| Chr7B[a] | 37,794,386 | 1 | 37,794,386 |
| Chr8A[a] | 41,957,870 | 1 | 41,957,870 |
| Chr8B | 40,861,194 | 2 | 24,818,166 |
| Chr9A | 58,551,545 | 5 | 29,408,779 |
| Chr9B[a] | 57,786,484 | 1 | 57,786,484 |
| UA | 19,868,619 | 314 | 93,022 |
| **Total** | **850,463,639** | **343** | **37,846,408** |

[a]Chromosomes with T2T assembly.

18 chromosomes (Supplementary Table 3). The estimated centromere sizes are provisional and likely include surrounding pericentromeric regions, as they were inferred from the distribution of centromere-associated tandem repeats (CentTR1 and CentTR2) and centromeric retrotransposons (CRMs). Most chromosomes exhibited a high density of centromeric repeats; however, Chr1A, Chr5A, and Chr9B showed a comparatively lower abundance (Supplementary Fig. 3), potentially due to unresolved assembly gaps or intrinsic structural variation. Telomeric regions were identified by detecting 7-bp repeat units (AAACCCT) within 100 kb of both chromosomal ends. This analysis revealed that eight of the 18 chromosomes contained telomeric repeats at both ends, eight chromosomes had repeats at only one end, and two chromosomes lacked identifiable telomeric repeats entirely (Supplementary Table 4). Additional gap closure, particularly using ultra-long reads, will be required for improving the resolution of both centromeric and telomeric regions.

### Repeat annotation and gene prediction

Repeat annotation revealed that 58.4% (496.4 Mb) of the assembled genome comprises repeat elements, including transposable elements (TEs; 55.6%) and tandem repeats (Supplementary Table 5). Among TEs, long terminal repeat (LTR) retrotransposons are predominant, accounting for 39.5% of the genome. The LTRs are classified into two main superfamilies: Gypsy (33.2%) and Copia (6.3%). Analysis of full-length LTR elements identified 5863 elements, with Gypsy elements (3762) significantly outnumbering Copia elements (1633) (Supplementary Table 6). Within Gypsy elements, the Gypsy-Tekay family was the most abundant, comprising 2209 copies, followed by the Gypsy-CRM family with 684 elements. Among Copia elements, the Ale family (426 copies) and the SIRE family (380 copies) were the most prevalent (Supplementary Table 6). Non-LTR retrotransposons and DNA transposons collectively made up 16.1% of the genome, while tandem repeats and non-coding RNA elements contribute an additional 2.8%. To evaluate the assembly quality of repetitive regions, the LTR Assembly Index (LAI) was calculated for LTR retrotransposons, yielding a score of 15.3 (Supplementary Table 7). This high LAI score indicates a highly contiguous assembly of the repetitive fraction of the genome.

To support the annotation of protein-coding genes, we incorporated previously published[12] RNA-seq data from JK-8 tissue samples collected across 10 distinct developmental stages covering three major growth phases: emergence [germinating seeds (GS), radicle (RD), plumule (PU)], vegetative [young leaf (YL), young root (YR), crown meristem (CM), vegetative stem (VS)], and reproductive [early panicle (PE), mid-panicle (PM), late panicle (PL)]. Using a comprehensive approach that combined ab initio gene prediction via Braker3, with homology-based evidence from the reference broomcorn millet protein database, and the RNA-seq transcripts of little millet, led to the identification of 59,045 non-redundant genes. Among these, 5996 genes (5.3%) contained two or more alternatively spliced isoforms. Over 99.99% (58,785) of the annotated genes were mapped to chromosomes, with only a small fraction remaining on unanchored scaffolds. To assess the quality of the genome assembly and gene annotation, we performed Benchmarking Universal Single-Copy Orthologs (BUSCO) analysis. Based on the embryophyta10 database, little millet achieved a 99.3% complete BUSCO score (Fig. 2A and Supplementary Fig. 4), while broomcorn millet scored 98.8%[14]. Furthermore, remapping the Illumina NGS reads (10.6 Gb) to the JK-8 genome assembly demonstrated a high mapping rate of 99.9% with uniform coverage across nearly all genomic regions. Together with the high LAI score (Supplementary Table 7), these results confirm the high quality of the de novo JK-8 assembly.

The analysis of transcription factors in the little millet genome revealed that approximately 5.9% (3503 out of 59,045) of the total genes encoded TFs, distributed across 56 distinct TF families (Supplementary Data 1 and 2 and Supplementary Fig. 5). Among these, the bHLH family was the largest, with 326 genes, followed by the ERF (297 genes), MYB (259 genes), and NAC (251 genes) families, all of which are essential for regulating gene expression during various biological processes. bHLH TFs are known to play a role in iron homeostasis in plants. Other notable TF families included C2H2 (231 genes), bZIP (188 genes), and WRKY (179 genes), which play key roles in stress responses and developmental processes. Families such as MYB-related (124 genes), GRAS (124 genes), and B3 (107 genes) are also important contributors to the gene regulatory network. Smaller but functionally important families include HD-ZIP (90 genes), MIKC_MADS (76 genes), and FAR1 (70 genes; Supplementary Data 2). InterProScan analysis identified 4468 distinct Pfam (protein families) domains across the 59,045 predicted genes. Among these, 28,548 genes contained a single Pfam domain, while 16,497 genes had two or more Pfam domains (Supplementary Data 3). The Pfam annotations indicated that the Protein Kinase domain (PF00069) was the most abundant, followed by Leucine-Rich Repeats (PF00560 and PF13855), Protein Tyrosine and Serine/Threonine Kinase domains (PF07714), PPR repeats (PF01535 and PF13041), Cytochrome P450 (PF00067), and the Ring Finger domain (PF13639), among others (Supplementary Data 3). In addition, Gene Ontology (GO) terms were assigned to 21,448 genes, KEGG Orthology (KO) and pathways to 15,175 genes, and Enzyme Commission (EC) numbers to 10,921 genes, enabling a broad understanding of biological processes, molecular functions, and pathway-level organization (Supplementary Data 4).

## Synteny and collinearity with other millet species

The identification of 1541 duplicated genes out of 1600 embryophyta BUSCOs (95.5%; Fig. 2A) suggested that the little millet genome is likely a polyploid. This was further supported by Smudgeplot analysis, which revealed patterns consistent with allotetraploidy (Fig. 2B). To confirm this, we used BLASTP to identify sequence homology between little millet proteins and the proteome of the closely related species broomcorn millet. Syntenic gene chains between two species were computed using DAGChainer, with broomcorn millet protein-coding genes serving as genomic anchors to identify corresponding syntenic orthologs (syntelogs) forming collinear conserved blocks in the little

millet genome. The matrix representing the syntenic relationships between broomcorn millet genes and their corresponding little millet homologues is provided in Supplementary Data 5. This analysis revealed that the duplicated chromosomes in little millet mirror the genomic structure of broomcorn millet and confirmed that little millet is a tetraploid (Fig. 2C and Supplementary Data 5). In total, 41,070 little millet genes were identified as syntenic orthologs of broomcorn millet genes, while the remaining 17,975 genes were classified as either tandem duplicates or non-syntenic genes. These syntelogs were further categorized based on their retention or loss: fully retained (both homeologs retained, 19,147 genes), partially fractionated (one homeolog lost, 2776 genes), or fully fractionated (both homeologs lost, 8367 genes) (Supplementary Data 5). These classifications offer valuable insights into the structural organization and evolutionary history of the little millet genome. For example, a comparative analysis of structural differences between the little millet and broomcorn millet sub-genomes revealed 16 and 11 inversions in little millet sub-genomes 1 and 2, respectively. Additionally, 21 inversions were identified in both little millet sub-genomes when compared to broomcorn millet (Fig. 2C and Supplementary Data 6). A reciprocal translocation was also detected between chromosomes Chr4A and Chr4B, and Chr5A and Chr5B. (Supplementary Data 6). The Hi-C interaction matrices for Chr4A, Chr4B, Chr5A, and Chr5B confirmed these reciprocal structural variations (Supplementary Fig. 2).

Gene expression levels play a crucial role in determining the fate of duplicated gene copies following whole-genome duplication[15,16]. To investigate sub-genome dominance in little millet, we analyzed RNA-seq samples[12] from ten distinct tissues covering emergence, vegetative, and reproductive stages (Fig. 2D). After removing genes expressed at a low level from fully retained homoeologous pairs, we assessed expression bias and classified homoeologous gene pairs into three categories: balanced, A-sub-genome-dominant, and B-sub-genome-dominant (Fig. 2D and Supplementary Data 6). Across different tissues, 3.5–9.7% of expressed homoeologous pairs were found to be A-dominant, while 3.4–9.8% were B-dominant (Fig. 2D). Furthermore, some gene pairs exhibited varying dominance across tissues and were classified as dynamic, whereas those with consistent dominance patterns were considered stable. Among all expressed homoeologous pairs, 79.0% were stable and 21.0% dynamic (Supplementary Fig. 6). These results suggest that expression bias in little millet is primarily gene- and tissue-specific, rather than indicative of broad genome-wide dominance.

A comparison of the little millet genome with those of other millet and related cereal species was conducted using a k-mer-based, reference-free panKmer analysis (Fig. 3A) alongside phylogenetic analysis (Supplementary Fig. 7). Both methods produced consistent results, showing that broomcorn millet is the closest relative to little millet. These two species were clustered together with other millet species, including barnyard grass (*Echinochloa crus-galli*), foxtail millets (*Setaria* spp.), pearl millet (*Pennisetum glaucum*), and fonio millet (*Digitaria exilis*). In contrast, rice (*Oryza sativa*), finger millet (*Eleucine coracana*), sorghum (*Sorghum bicolor*), and maize (*Zea mays*) formed distinct clades separate from the primary millet group. Additionally, Arabidopsis, Medicago, and quinoa were placed in a completely separate clade, consistent with their status as outgroup species. To estimate the relative age of divergence between little millet and other millet and cereal species, the Bayesian method MCMCTree was employed, providing a robust evolutionary framework. This analysis revealed that little millet and its closest relative, broomcorn millet, diverged approximately 9.05 million years ago (Mya; Fig. 3B). All millet species were inferred to have diverged from maize and sorghum approximately 28.56 Mya, with subsequent speciation events (Fig. 3B). These findings support the evolutionary relationships among millet species[12], and highlight the close genetic relationship of little millet

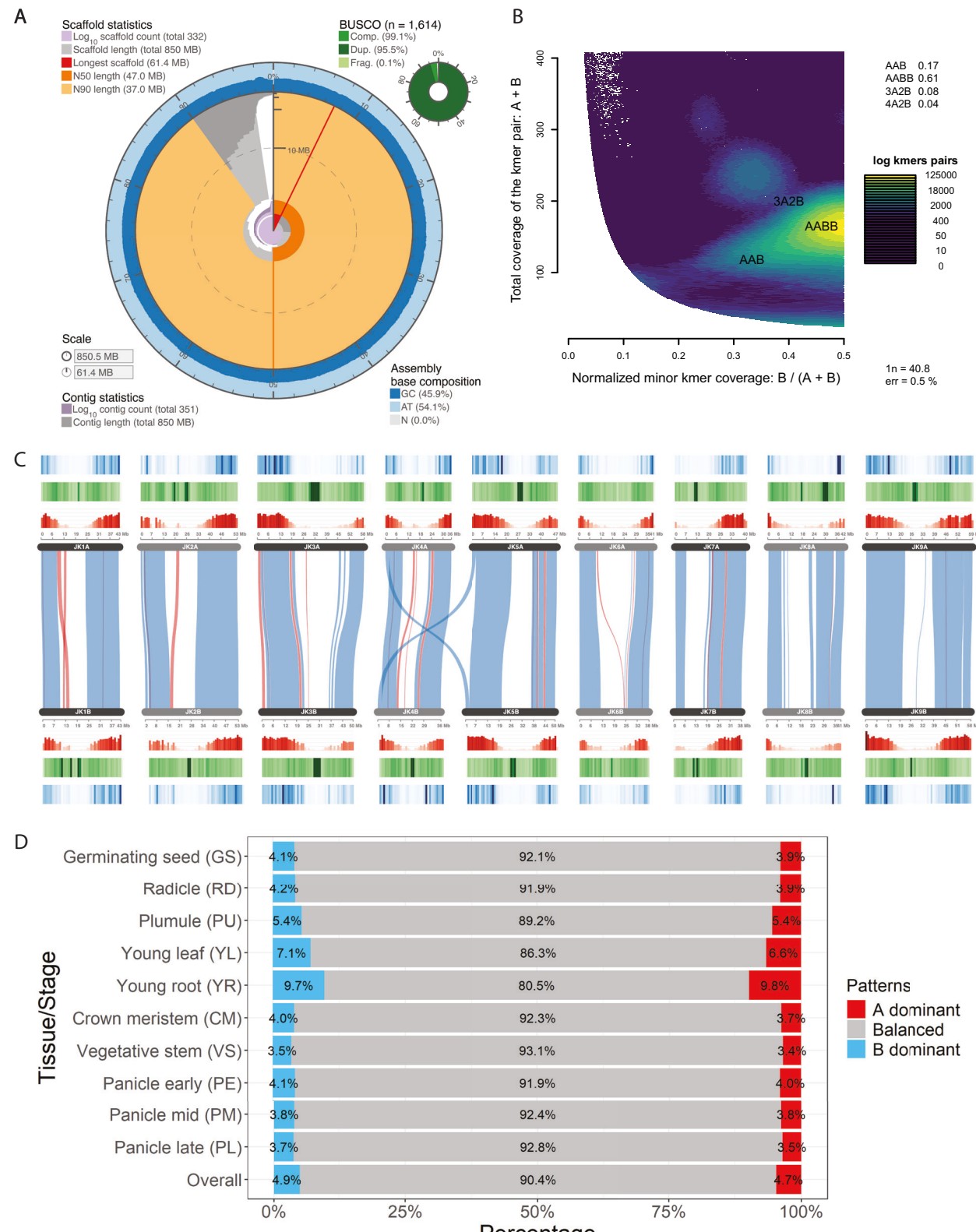

with broomcorn millet and greater divergence from other millet species.

An analysis of the $Ks$ values distribution among the coding regions of homeologous gene pairs in little millet provided an estimate of the divergence time between the progenitors of its two sub-genomes. Mixture model analysis of the $Ks$ distribution identified a prominent peak at $Ks \approx 0.048$ (Fig. 3C). Using mutation rates calculated based on

the synonymous substitution ($Ks = 0.57038$; Supplementary Data 7) associated with the divergence between *Oryza sativa* and *Panicum miliaceum*, which occurred approximately 41.4–51.8 million years ago[17], the separation of the progenitors of the two sub-genomes of little millet was estimated to have occurred approximately 3.39 to 4.25 million years ago. This timing is more recent than the divergence of the progenitors of the two sub-genomes of broomcorn millet (4.35 to

**Fig. 2 | Genomic features and structural insights into the little millet genome.** **A** Snail plot displaying genomic features. Light gray color ($\log_{10}$ scale) at the center indicates scaffold counts, while dark gray marks scaffold lengths. The longest scaffold is shown as a red line, with its assembly coverage percentage in red. N50 and N90 metrics appear as dark and light orange blocks, respectively. GC and AT contents are displayed as dark and light blocks on the outer ring, with the total assembly length on the outermost scale. BUSCO completeness is shown in the top-right corner. **B** Smudgeplot analysis of the little millet genome reveals patterns consistent with allotetraploidy. **C** SynVisio plot visualizes genomic collinearity between the two sub-genomes A and B of little millet through linked ribbons that connect conserved regions. Regular syntenic regions are shown in blue while inverted regions are shown in red. Additionally, histogram and heatmap tracks have also been added to visualize gene density (red), repeat density (green) and gene expression values (blue). **D** Sub-genome dominance across tissues and developmental stages. This bar chart displays the proportions of significantly dominant gene pairs between sub-genomes A and B across various tissues. The *x*-axis shows percentages of dominance patterns; the *y*-axis lists tissues from emergence, vegetative, and reproductive phases. The overall percentage across all stages is shown at the bottom. Color codes for dominance patterns are indicated in the right panel. Source data are provided as a Source Data file.

5.45 Mya; Fig. 3C and Supplementary Data 7). These findings suggest that polyploidization in both little millet and broomcorn millet likely occurred within the last 5.45 million years. Additionally, genetic differentiation analysis between the sub-genomes of little millet and broomcorn millet revealed that the duplicated genomes within each species are more similar to each other than to those of the other species (Fig. 3D). Furthermore, the highly undifferentiated nature of the sub-genomes in both species indicates that their polyploidy events are relatively recent and occurred after the speciation of these two species (Fig. 3E).

A comparative analysis of repeats between little millet and broomcorn millet revealed distinct patterns of genome evolution. Little millet has a higher repeat content (58.55%) compared to broomcorn millet (52.3%; Supplementary Table 5). Long Terminal Repeat (LTR) retrotransposons dominate in both genomes, with similar numbers of full-length LTRs (5863 in little millet vs. 5870 in broomcorn millet; Fig. 3F). However, Gypsy-Tekay elements are abundant in little millet (2209 copies), while they are scarce in broomcorn millet (153 copies), contributing to genome expansion and structural diversity. Conversely, Gypsy-Ogre elements are much more prevalent in broomcorn millet (2090 copies vs. 27 in little millet), playing a key role in genome stability. Age distribution analysis revealed a burst of Gypsy-Tekay activity in little millet within the past million years, while Gypsy-Ogre elements in broomcorn millet have been active in the past two million years (Supplementary Fig. 8). These differences highlight divergent evolutionary pressures, with Gypsy-Tekay driving dynamism in little millet and Gypsy-Ogre maintaining stability in broomcorn millet. Gypsy-Ogre elements are evolutionarily conserved across a wide range of plant taxa and exhibit stress-responsive behavior, with their expression modulated by abiotic stressors such as heat and oxidative stress[18–20]. These elements are also major components of legume genomes and may contribute to genetic diversity and adaptation to environmental pressures[18,21]. Together, these findings highlight the role of Gypsy-Ogre elements in promoting genome stability under stress, while emphasizing how lineage-specific evolutionary pressures may have shaped the distinct repeat landscapes of little millet and broomcorn millet.

### Genetic variation and population structure of little millet
To assess genetic diversity and analyze the population structure of little millet, a panel of 300 diverse accessions (Supplementary Data 8) including breeding lines and germplasm collections from ten geographically distinct regions of India was selected for comprehensive genomic analysis. These accessions were sequenced using Illumina technology generating over 2.14 trillion base pairs of high-quality short reads (Supplementary Data 9). This resulted in an average sequencing depth of 8.44X when mapped to the JK-8 reference genome (Supplementary Data 9). The raw reads were processed to remove adapter contamination, PCR duplicates, ambiguous residues (N's), and low-quality regions before alignment to the JK-8 reference genome. From the alignments, a total of 739,712 single nucleotide polymorphisms (SNPs) were identified across the 300 accessions. After rigorous filtering, 249,511 high-quality SNPs were retained for population genetic

analyses, yielding an average density of 293.3 high-quality SNPs per megabase of the genome (Supplementary Data 9).

The neighbor-joining phylogenetic tree based on the 249,511 SNPs identified four major clades within the diversity panel (Supplementary Fig. 9A). Population structure analysis using fastSTRUCTURE revealed that the marginal likelihood plateaued at $K = 10$, indicating ten distinct sub-populations (Supplementary Fig. 9B). ADMIXTURE identified $K = 11$ as the optimal number of ancestral populations, whereas both DAPC and STRUCTURE supported $K = 9$ clusters (Supplementary Fig. 10). These results aligned with fastSTRUCTURE, collectively indicating a well-defined population structure among the accessions. However, the sub-populations from these analyses did not align with geographic origins (Supplementary Data 10), as each group included accessions from multiple regions across India. This pattern likely reflects the limited genetic diversity within the little millet germplasm, frequent exchange of breeding materials among regional programs, and recurrent use of elite parental lines selected for a narrow range of agronomic traits. Some advanced breeding lines were also derived from common ancestral sources, further contributing to genetic similarity across distant locations. These population clusters were further validated by principal component analysis (PCA) of the SNP dataset (Fig. 4A). Linkage disequilibrium (LD) in the JK-8 reference genome varied from 0.6 Mb on Chromosome 1B to 8.1 Mb on Chromosome 6B (Supplementary Fig. 11). As a highly self-pollinated crop, little millet exhibits significantly higher LD values compared to crops like pearl millet, soybean, and pea, suggesting a stronger genetic cohesion within its germplasm. These differences in LD decay are important for interpreting the resolution of GWAS signals, as regions with slower decay may limit fine-mapping resolution, while those with faster decay can provide higher precision in identifying candidate loci.

### Structural variant discovery, validation, and distribution
Structural variants (SVs) were identified across the 300-accession diversity panel using a dual-caller approach combining Delly and Manta. SV calling was performed independently for each accession using both tools. The SVs detected by both callers were merged using SURVIVOR, resulting in a high confidence consensus set of 14,273 SVs (Supplementary Fig. 12).

This consensus set was dominated by deletions (11,422), followed by duplications (1223), inversions (898), and insertions (730) (Supplementary Fig. 12 and Supplementary Table 8). Among the deletions, 4252 (29%) fell within the 30–50 bp size range (Fig. 4B and Supplementary Table 8), and overall, 73% of the SVs were shorter than 1000 bp, reflecting the prevalence of small structural variants in the panel. Translocations were excluded due to limitations in accurate detection and validation using short-read data.

To validate SV calls, we applied a read-depth-based approach using Samplot, which provided visual confirmation of structural events through characteristic coverage patterns. Representative examples of high-confidence deletions and duplications showed strong support, reinforcing the reliability of the consensus SV set (Supplementary Fig. 13). SVs were unevenly distributed across the genome, with chromosome 8A containing the highest number (1225) and chromosome

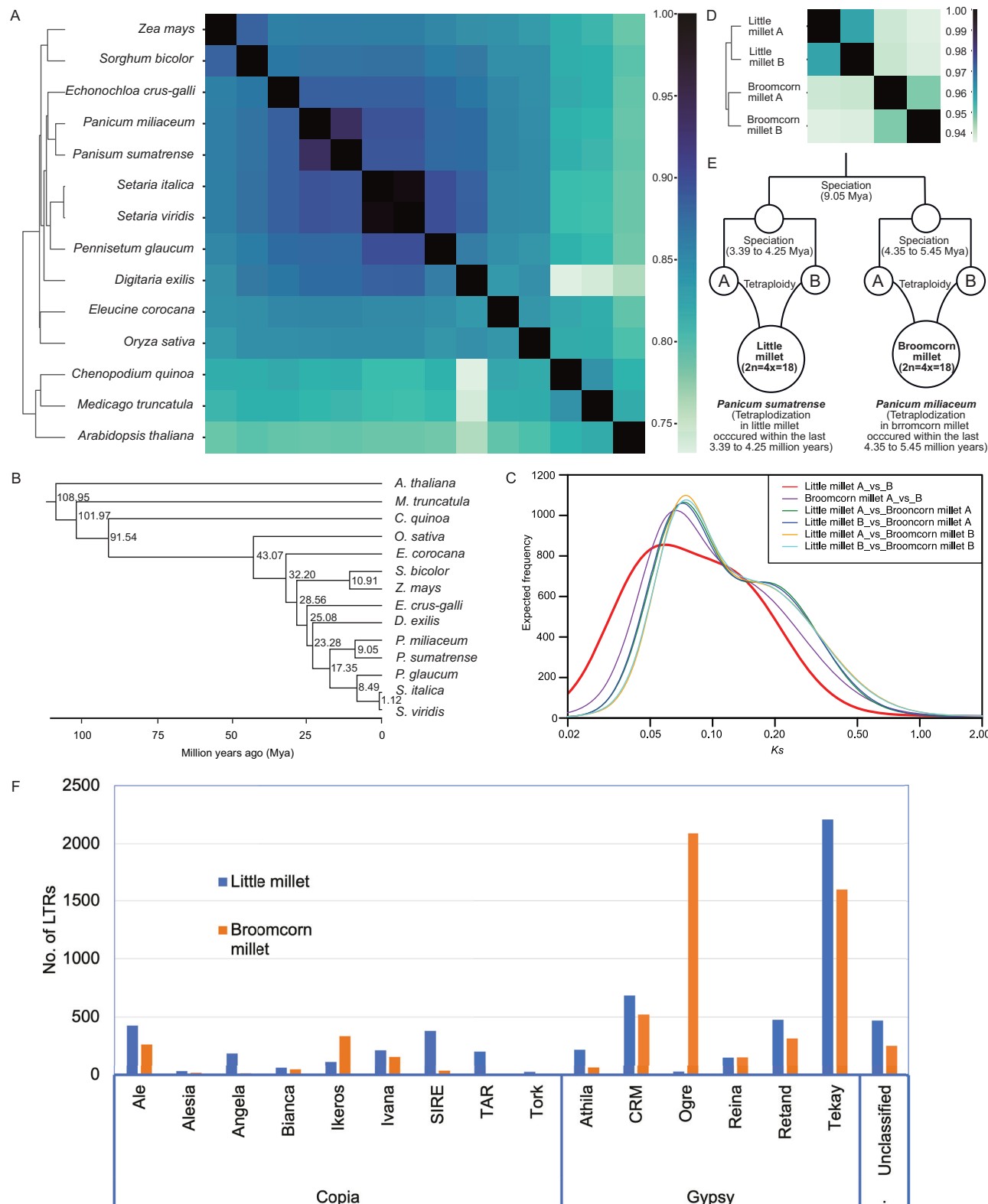

6B the lowest (719), indicating variable structural variation across chromosomes (Supplementary Fig. 14 and Supplementary Table 9).

Deletions were primarily intergenic, with 49.8% located within 3 kb from genes, 12.9% within genic regions, and 17.0% within 1 kb of genes (Supplementary Fig. 15 and Supplementary Table 10). In contrast, duplications and inversions exhibited more genic proximity, with 65.3% and 69.6%, respectively, located within 3 kb of genes. Insertions followed a similar trend, with 60.4% positioned near gene regions.

These patterns suggest that different classes of SVs exhibit distinct spatial relationships with coding regions, potentially influencing their functional impact.

We utilized LSV-viz (see "Methods") to identify large structural variants (LSVs; > 40 Kb) within the little millet population by visualizing entire chromosome line plots (Supplementary Figs. 16–18). This approach enabled the detection of LSVs across all chromosomes, with their precise locations manually extracted from the raw LSV

**Fig. 3 | Genetic relationships between little millet and other millet species. A** A heatmap representation of shared gene families (pankmer analysis) between little millet, other millet species, and related cereal species. The intensity of heatmap coloration indicates the degree of genetic similarity, with darker regions representing a higher proportion of shared genes. Little millet shows a high degree of similarity with broomcorn millet and other millet species but relatively lower overlap with distantly related cereals, reflecting phylogenetic divergence. **B** Divergence time estimates among millets and related cereals. A species tree was constructed from single-copy orthologs across 14 plant genomes using IQ-TREE.

Divergence times were estimated with MCMCtree under an independent rates model. **C** Synonymous substitution rate (*Ks*) distributions between different sub-genomes of little millet and broomcorn millet. **D** Genetic relationships between the sub-genomes of little millet and broomcorn millet. The figure displays the output of a PanKmer analysis comparing the sub-genomes of little millet and broomcorn millet. **E** An evolutionary model depicting the speciation event between little millet and broomcorn millet occurred before the genome duplication in each species. **F** Distribution of various LTR elements in the genomes of little millet and broomcorn millet. Source data are provided as a Source Data file.

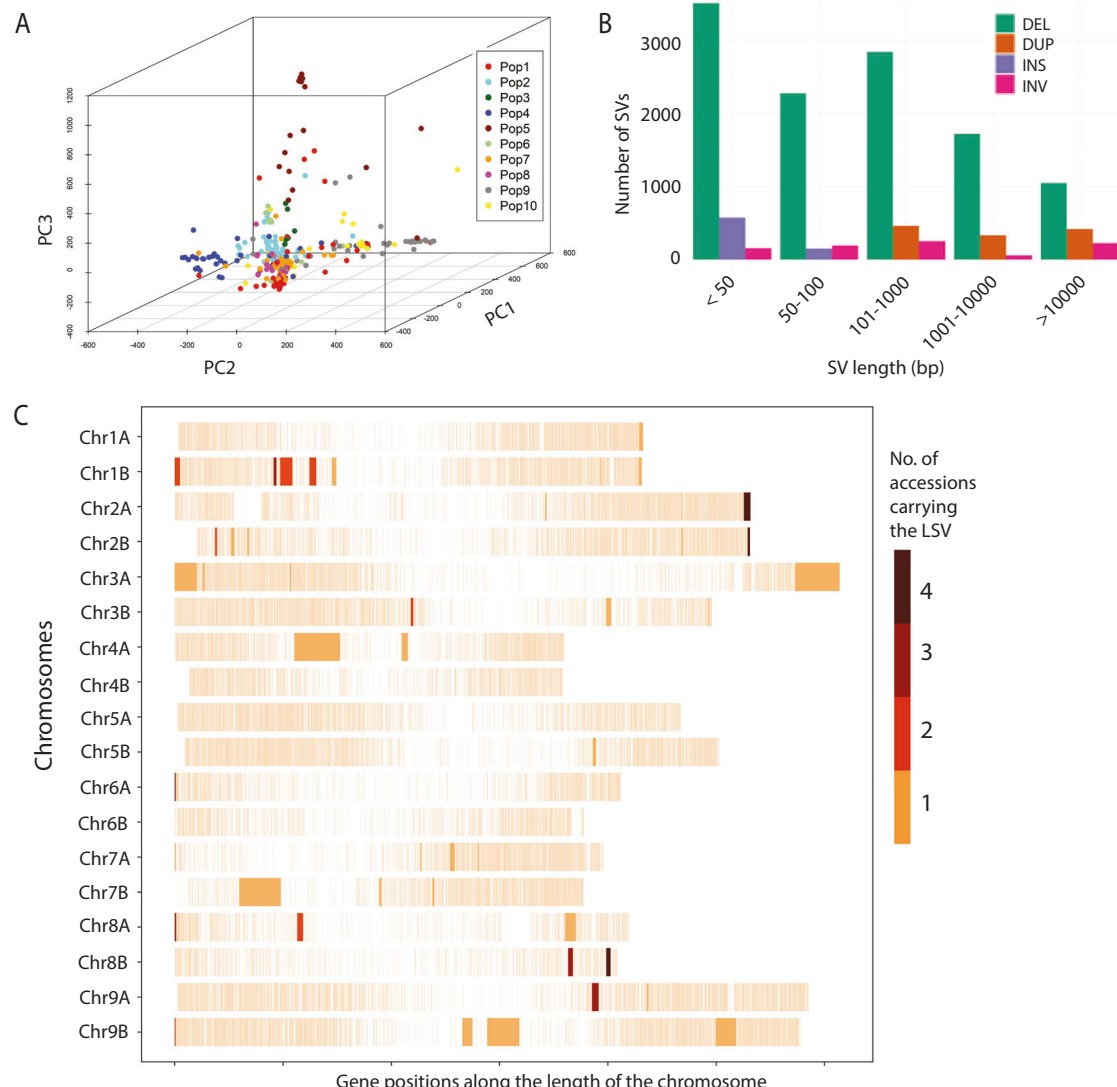

**Fig. 4 | Genetic variation in the little millet diversity panel. A** Three-dimensional PCA plot of diversity panel based on the first three principal components. The accessions were color coded to align with the populations identified in structure analysis in this figure (**B**). **B** Size distribution of structural variants (SVs) identified in the diversity panel. **C** Distribution of large structural variants (LSVs; >40 Kb) in the diversity panel. Large structural variants from the little millet diversity panel were identified and mapped onto their respective chromosomal locations. Gene density is shown as lighter marks along the lengths of the chromosomes. The large

structural variants appear in solid colors based on the number of lines in the diversity panel that have a large structural variation at the exact same location. Note that the ends of the chromosomes are not explicitly marked except for occurrences of large structural variants that extend to the ends of the chromosomes. As a result, there current boundaries of the chromosomes on this plot are indicated by the positions of the first and last gene on the chromosome. Source data are provided as a Source Data file.

data output. A total of 47 LSVs were identified, distributed across 15 chromosomes (Fig. 4C and Supplementary Figs. 19 and 20). These variants ranged in size from 4 to 329 genes, with an average of 35 genes per LSV. The length of the identified LSVs varied from 40,308 base pairs to 4,217,303 base pairs, with an average length of 568,567

base pairs. Seven of these LSVs were larger than one megabase in size, with an average length of 2,435,283 base pairs and containing an average of 126 genes per LSV. In total, these 47 LSVs encompassed 1670 genes, which were distributed across the identified variants. A summary of the identified LSVs, including their chromosome

locations and corresponding gene counts, is presented in Supplementary Data 11.

## Field phenotyping for agronomic traits and grain mineral composition

The diversity panel was assessed for agronomic traits and grain micronutrient concentration at two locations, the ICAR-Indian Institute of Millets Research (IIMR), Hyderabad, India and Gandhi Krishi Vigyana Kendra (GKVK), Bangalore, India, using an alpha lattice design. A representative example of the panicle color diversity in the panel is shown in Fig. 5A. Three agronomic traits, including days to flowering (DFF), thousand seed weight (TW) and grain yield per plant (GYPP), along with seven grain micronutrients (Ca, Fe, K, Mg, Na, P, and Zn) were measured in the diversity panel, which in general exhibited an approximately normal distribution (Fig. 5B). However, GYPP, Fe, Na and K displayed right-skewed distributions, while TW exhibited a left-skewed distribution.

The agronomic traits studied are important for large-scale crop adoption. Our analysis revealed significant diversity in the panel for these traits. Statistically significant ($P < 0.01$) differences were observed both among accessions and locations (Table 2). These traits did not show significant variation across experimental replicates and blocks, confirming that randomization was effectively implemented. The BLUPs calculated from pooled data across both locations (Supplementary Data 12) indicated that DFF ranged from 44.5 to 77.5 days, with a median of 62.2 days (Fig. 5B). The variation in flowering time is critical for developing little millet cultivars suited to different geographical regions and climatic conditions. TW varied from 2.1 to 3.6 × $g$ and GYPP ranged from 3.7 to 13.2 × $g$, supporting the suitability of the diversity panel for association mapping and selecting donors for trait

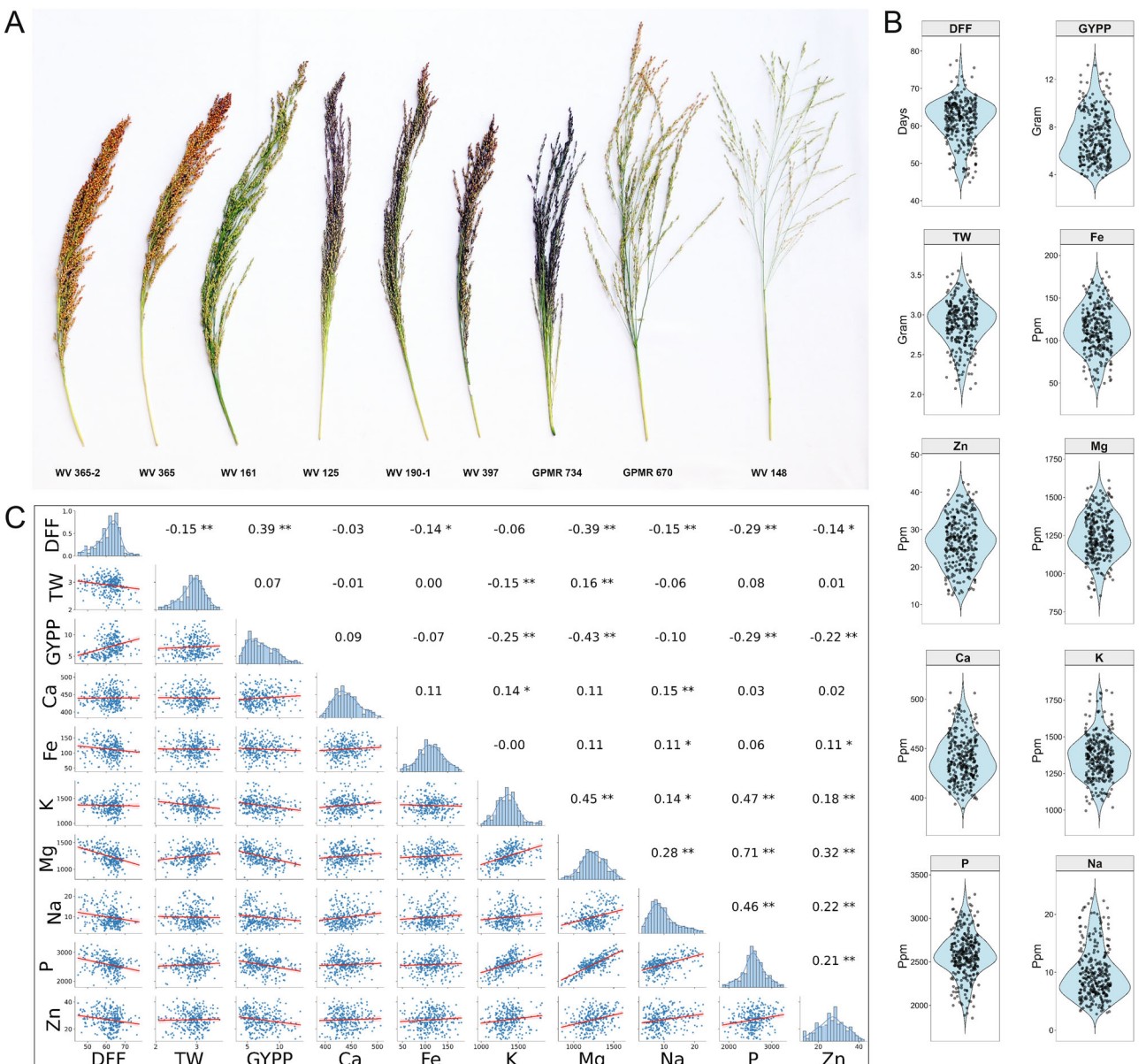

**Fig. 5 | Morphological and phenotypic variation in the little millet diversity panel. A** Morphological diversity of spikelets within the diversity panel. **B** Distribution of three agronomic (days to 50% flowering, DFF; thousand seed weight, TW; grain yield per plant, GYPP) and seven grain micronutrient concentrations in the diversity panel. The violin plots represent the BLUPs calculated based on testing the accessions in three replications and at two locations. **C** Spearman's rank correlation coefficients among agronomic traits and grain micronutrient concentrations were calculated using BLUP values derived from measurements at two test locations. Two-sided tests were performed and asterisks indicate significance levels (* $p < 0.05$; ** $p < 0.01$).

**Table 2 | Analysis of variance for agronomic traits and grain mineral concentrations of little millet diversity panel**

| Traits | Source of variation | | | | |
|---|---|---|---|---|---|
| | Genotype | Location | Replication | Block | Heritability |
| Days to flowering | $7.87^{E-265}$*** | $0$*** | $0.497^{ns}$ | $0.385^{ns}$ | 92.54% |
| Thousand seed weight | $4.63^{E-90}$*** | $1.93^{E-26}$*** | $0.366^{ns}$ | $0.909^{ns}$ | 91.94% |
| Grain yield per plant | $2.64^{E-94}$*** | $3.63^{E-11}$*** | $0.292^{ns}$ | $0.739^{ns}$ | 92.28% |
| Ca | $1.19^{E-15}$*** | $4.88^{E-260}$*** | $0.888^{ns}$ | $0.598^{ns}$ | 74.16% |
| Na | $5.47^{E-58}$*** | $1.79^{E-27}$*** | $0.764^{ns}$ | $0.447^{ns}$ | 89.00% |
| Zn | $0$*** | $7.27^{E-107}$*** | $0.430^{ns}$ | $0.400^{ns}$ | 99.47% |
| K | $5.74^{E-89}$*** | $1.63^{E-277}$*** | $0.559^{ns}$ | $0.695^{ns}$ | 91.97% |
| P | $1.38^{E-78}$*** | $3.68^{E-26}$*** | $0.381^{ns}$ | $0.846^{ns}$ | 94.28% |
| Fe | $6.85^{E-271}$*** | $9.01^{E-81}$*** | $0.464^{ns}$ | $0.010^{ns}$ | 93.86% |
| Mg | $3.62^{E-72}$*** | $6.34^{E-23}$*** | $0.378^{ns}$ | $0.957^{ns}$ | 90.46% |

Ca, Na, Zn, K, P, Fe, and Mg are seed micronutrients measured in ppm.

*, ** and *** denote statistically significant differences at P-value of <0.05, <0.01 and <0.0001, respectively. ns denotes non-significant differences (P-value > 0.05).

Statistical analysis was performed using a pooled analysis of variance (ANOVA) to assess the effects of genotype, location, replication and block on agronomic and grain micronutrient traits. Two-sided tests were used to determine significance levels. No adjustments for multiple comparisons were applied.

improvement. The heritability estimates for these agronomic traits were 92.5%, 91.9% and 92.3% for DFF, TW and GYPP, respectively (Table 2).

The grain micronutrient concentration within the diversity panel accessions varied from 30.1% to 90.1% for Ca, K, Mg, and P, 240.4% for Zn, 301% for Fe, and 717.5% for Na (Fig. 5B and Supplementary Data 12). Similar to agronomic traits, significant differences (P < 0.01) were observed in grain micronutrient concentration across accessions and test locations. However, the heritability of these micronutrients varied considerably. The heritability values were 74.2% for Ca, 89.0% for Na, 99.5% for Zn, 92.0% for K, 94.3% for P, 93.9% for Fe and 91.5% for Mg, highlighting the considerable impact of environmental factors, such as crop growing conditions, maturity seasons, and soil quality on grain micronutrient concentration.

The grain micronutrient concentration of Ca, Fe, K, Mg, Na, P, and Zn, showed strong positive correlation with one another across both locations (Fig. 5C), suggesting that these nutrients tend to accumulate together under favorable conditions, leading to higher concentrations in the grain. K, Mg, Na, P, and Zn exhibited strong positive correlation among themselves, while Ca and Fe showed weak correlation (less than 0.2) with other micronutrients. Interestingly, the micronutrient concentration of Zn, P, Na, and Mg exhibited negative correlation with DFF and GYPP (Fig. 5C). GYPP showed a moderately strong positive correlation with DFF, indicating that a longer vegetative phase tends to favor higher grain yield. Similarly, TW was not strongly correlated with GYPP or micronutrient concentration, except for a negative correlation with K and DFF and positive correlation with Mg (Fig. 5C).

The diversity panel demonstrated considerable potential for enhancing grain micronutrient concentration and improving agronomic performance. Genotypes such as BL 6 (180 ppm) and WV-314-1 (175 ppm) exhibited high iron (Fe) content, while WV-260-1 (156 ppm) and WV-379 (155 ppm) stood out for their exceptional performance in both grain Fe and yield. Similarly, WV-169 (39 ppm) and WV-433 (39 ppm) showcased high zinc (Zn) content alongside high grain yield. IC-0483133 and GPMR-734 were identified as elite candidates for biofortification and breeding programs due to their high levels of both Fe and Zn. The co-occurrence of elevated Fe and Zn in IC-0483133 and

GPMR-734 suggests a shared regulatory mechanism or closely linked loci, offering a promising avenue for developing biofortified varieties. Additionally, GPMR-660 emerged as a standout genotype, with high concentrations of Mg (1527 ppm), Na (20 ppm), P (3144 ppm), and Zn (37 ppm), combined with early flowering (50 days). These attributes make GPMR-660 a valuable candidate for breeding nutrient-rich and early-maturing cultivars.

## Genome-wide association (GWAS) mapping for agronomic traits and grain micronutrient composition

GWAS analysis was performed for DFF, TW and GYPP measured in the field and concentration of six grain micronutrients (Ca, Fe, K, Mg, Na, P, and Zn) measured post-harvest. The analysis utilized 249,511 SNPs, and SVs distributed across the 18 chromosomes of little millet. Phenotypic values from each test location were used both separately and as combined BLUP values for association analysis. The 300 accessions used in this study provide a sufficiently large sample size to identify genes and genomic regions associated with each of the analyzed traits. For SNP-based association analysis, two GWAS mapping models namely, MLMM and BLINK were applied. CMLM and BLINK models were used for SV-based association analysis.

The Bonferroni-based threshold for the SNP-based association mapping in this study is $-\log10(P) \approx 6.7$ (adjusted P-value < 0.05/249,511). To capture all potential true associations contributed by multiple minor effect loci below this threshold level, we employed a strategy similar to Deng et al. [22], applying a false discovery rate cutoff of $-\log10 (P = 10^{-4})$ to identify a list of candidate SNPs. These SNPs were further refined by comparing the data sets from the two locations and the pooled data, as well as between the association analysis models used, to identify candidate genes. The presence of candidate genes associated with metabolic pathways related to the micronutrients served as the final filter to confirm marker-trait associations related to micronutrient concentration. Numerous significant marker-trait associations with P-value < 0.0001 were identified across the traits, test locations, and pooled data (Fig. 6A, B, Supplementary Data 13 and Supplementary Fig. 21). Significant SNPs detected through GWAS remained largely consistent across all four population structure correction methods, fastSTRUCTURE (Supplementary Data 13), ADMIXTURE (Supplementary Data 14), DAPC (Supplementary Data 15) and STRUCTURE (Supplementary Data 16), reinforcing the robustness and reliability of the association results. Highly significant marker-trait associations were also identified for TW, GYPP, and P using the pooled phenotypic data from both locations. Notable markers identified in all four populations structure correction methods include Chr2A_3693904 (TW; P-value = 3.8e$^{-10}$), Chr4B_27491672 (GYPP; P-value = 7.61e$^{-08}$), and Chr1B_39041529 (P; P-value - 2.3e$^{-09}$). Out of the 145 SNPs selected across the phenotypic data sets and models, potential candidate genes associated with micronutrient pathways within 100 kb flanking regions were explored. Using fastStructure-based correction method, the flanking regions of two SNPs Chr1A_560273 and Chr2A_32889668 were found to contain genes associated with Fe metabolic pathways. The SNP Chr2A_32889668 showed the strongest association with Fe concentration among all the 249,511 markers when analyzed using BLINK on pooled data from two locations, with a P-value of 4.93e$^{-06}$. Similarly, the SNP Chr1A_560273 was a significant marker for Fe concentration, with a P-value of 8.71e$^{-06}$. The trait-association of these SNP markers was further confirmed by their consistent significance in GWAS based on ADMIXTURE, STRUCTURE and DAPC-corrected structure grouping (Supplementary Data 17). GWAS incorporating population structure inferred from fastSTRUCTURE, STRUCTURE, DAPC, and ADMIXTURE revealed a high degree of concordance in the identified significant SNPs, with only marginal differences in P-values. This consistency across models underscores the robustness of the detected marker-trait associations.

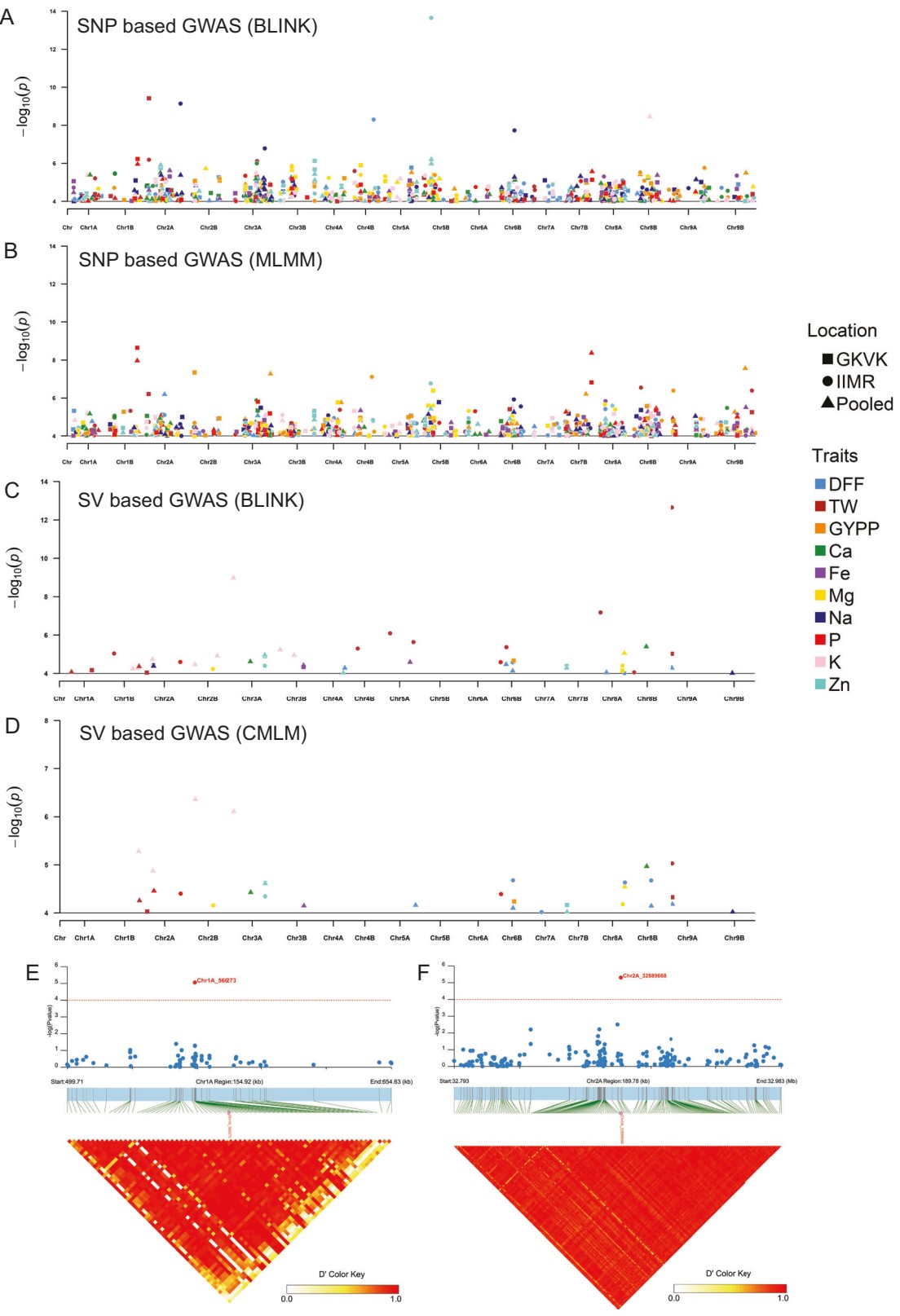

Haplotype analysis of significant MTAs associated with Fe, GYPP, TW and P revealed well-defined haplotype groups exhibiting significant differences in trait expression (Supplementary Fig. 22). These findings highlight the contribution of specific haplotype groups to phenotypic variability and suggest their potential utility in trait-based selection and breeding strategies.

The SV-based association analysis identified 84 significant SVs associated with DFF, GYPP, TW and all the seven seed mineral concentrations—Ca, Mg, K, P, Na, Fe and Zn across IIMR and GKVK test sites, and pooled data (Fig. 6C, D, Supplementary Data 18, and Supplementary Fig. 23). These significant markers include an SV on chromosome 8B (Chr8B_19025292), associated with Ca concentration in

**Fig. 6 | Genome-wide association study of agronomic traits and seed micro-nutrient levels.** Distribution of trait-associated SNPs across the genome determined by BLINK (**A**) and MLMM (**B**) models. Similarly, the distribution of trait-associated structural variants across the genome determined by BLINK (**C**) and CMLM models (**D**). The Manhattan plots display SNPs and structural variants associated with all tested traits at a significance threshold of $p > 10^{-4}$. Data were derived from individual test locations (GKVK, Bangalore, and ICAR-IIMR, Hyderabad, India) as well as BLUPs calculated based on pooled data from both locations. The tested traits include days to 50% flowering (DFF), thousand seed weight (TW), grain yield per plant (GYPP), and seed concentrations of Ca, Fe, Mg, Na, P, K, and Zn. Genome-wide association analyses were performed using the BLINK and MLMM models implemented in GAPIT. Two-sided tests were applied to assess marker-trait associations and $p < 0.0001$ were considered significant. Heatmaps illustrate the linkage disequilibrium (LD) blocks spanning 100 kb flanking regions of Fe-associated SNP markers Chr1A_560273 (**E**) and Chr2A_32889668 (**F**). These SNP markers, identified through GWAS analysis, are located near known genes involved in the Fe metabolic pathway. LD decay was estimated based on pairwise squared correlation coefficients ($r^2$) between SNP markers using a two-sided test.

both BLINK and CMLM models, linked to a 28.7 Kb deletion. For Fe concentration, an SV on chromosome 3B (Chr3B_34791532) explained 10.6% of the variation. Mg concentration showed the highest association with an SV on chromosome 8A (Chr8A_35716302) across both models, while Na concentration was associated with an SV on chromosome 9B (Chr9B_26265196). For TW, SV associations included Chr9A_7648531 and Chr1A_37424986 which explained phenotypic variation of up to 16.2%. Additionally, a few traits were located within 2 Mb of significantly associated SNP markers, for example SV marker Chr2A_1933725 and SNP marker Chr2A_3693904 associated with TW (Supplementary Data 17), enhancing confidence in the identified markers for these traits.

To investigate potential phenotypic associations of LSVs, an interquartile range (IQR) outlier analysis was performed on the phenotypic data of accessions carrying LSVs (Supplementary Data 19). This analysis aimed to identify any lines with LSVs that were phenotypic outliers across three phenotypic datasets (GKVK, IIMR, and pooled). A total of 28 accessions containing LSVs were found to be outliers for several agronomic and micronutrient traits (Supplementary Data 19). One such accession, WV-356-1, which carries a homozygous deletion in the telomeric region of chromosome 6A (Supplementary Fig. 24A), showed a significant advantage in Fe accumulation (399 ppm, compared to the average of 133 ppm across all 300 lines, Supplementary Fig. 24B). This relatively small deletion includes four genes. Although the exact roles of these genes in Fe metabolism are unknown, future research involving gene editing or overexpression could help elucidate their function.

## Genes involved in iron uptake and mobilization

Analysis of ionome-related genes[23] in little millet revealed significant enrichment compared to rice and Arabidopsis, with some genes present in as many as ten copies corresponding to a single gene in either species (Supplementary Data 20). A total of 185 micronutrient-related genes were identified, including 67 linked to Fe, and 44 to Zn. To assess the reliability of these high copy number estimates and to rule out potential assembly artifacts, we examined the mapping depth of ionome-related genes in JK-8 and a few other cultivars. This analysis confirmed consistent and elevated read depth across these loci, supporting their high-copy status and effectively ruling out assembly errors. The high copy numbers of Fe- and Zn-associated genes suggest robust mechanisms for acquiring and regulating these nutrients, contributing to little millet's adaptability to nutrient-poor environments.

Iron is a critical micronutrient for human health, with its deficiency causing anemia. Little millet is a rich source of Fe, which is absorbed through two strategies: Fe reduction and Fe chelation, depending on soil conditions. In Fe reduction, ferric chelate reductase (EC.1.16.1.7) converts $Fe^{3+}$ to $Fe^{2+}$ at the rhizosphere. Twenty reductase genes were identified in the JK-8 genome, including Psum.JK8.1r.7Ag009450 and Psum.JK8.1r.7Bg009660, homologous to *Os-FRO1* and *Os-FRO2* in rice and *At-FRO2* in Arabidopsis. Soluble $Fe^{2+}$ is absorbed via iron-regulated transporters like Psum.JK8.1r.9Ag012870, Psum.JK8.1r.9Ag012860, and Psum.JK8.1r.9Bg012780, homologous to *Os-IRT1*, *At-IRT1*, and *At-IRT2*. In Fe chelation, Poaceae species secrete 2′-deoxymugineic acid (DMA) into the rhizosphere via TOM1

transporters. DMA binds $Fe^{3+}$ in calcareous soils, forming $Fe^{3+}$-DMA complexes absorbed by YSL/YS1 transporters (Psum.JK8.1-r.1Ag022220, Psum.JK8.1r.1Bg022030). DMA synthesis involves nicotianamine synthase (Psum.JK8.1r.2Bg039160), nicotianamine aminotransferase (Psum.JK8.1r.1Ag012050), and deoxymugineic acid synthase (DMAS). This pathway occurs in root vesicles and follows a daily rhythm of phytosiderophore secretion. Figure 7 illustrates the little millet orthologs of rice and Arabidopsis genes involved in the DMA biosynthesis pathway.

Soil properties, including pH and oxygen levels, largely determine the preferred iron uptake strategy[24]. Iron homeostasis is regulated by bHLH TFs, conserved across grass and non-grass species[25]. In JK-8, homologous bHLH TFs, including Psum.JK8.1r.7Ag005670, Psum.JK8.1r.7Bg005810, Psum.JK8.1r.5Bg016540, Psum.JK8.1r.4Bg001350, Psum.JK8.1r.9Bg034120, Psum.JK8.1r.6Ag002520, and Psum.JK8.1-r.1Ag001050, form a regulatory network essential for Fe uptake, transport, and storage (Fig. 7).

Comparison of GWAS SNPs, SVs, and LSV-associated genes with ionome pathway genes identified 11 genes involved in multiple micronutrient pathways (Table 3). Among these, 5 genes are associated with Fe pathways: Psum.JK8.1r.3Ag000590 (ZIP2), Psum.JK8.1-r.3Ag002970 (MYB2), Psum.JK8.1r.3Ag002980 (MYB2), Psum.JK8.1r.9Bg041540 (SPT), and Psum.JK8.1r.9Bg042380 (DMAS1), along with Psum.JK8.1r.2Ag013440 (CPS4). These genes present opportunities for functional validation through overexpression or knockout studies to elucidate their roles in iron metabolism. Harnessing this knowledge could aid in developing biofortified little millet varieties with higher iron content.

## Discussion

Global food security heavily relies on a few staple crops, including rice, wheat, maize, and soybean, which together account for more than 60% of the world's caloric intake. However, this dependency poses significant risks to food security, as a widespread disease outbreak or climate-change-related event impacting any of these key crops could have devastating consequences. Beyond the risk of food security, these staple crops are also nutritionally limited, often deficient in essential micronutrients like iron and zinc, as well as dietary fiber. Millets provide a compelling alternative to mitigate these risks. They are high in dietary fiber, and antioxidants, nutritionally dense, rich in micronutrients such as calcium, iron, and zinc. In addition, millets are climate-resilient and capable of thriving on marginal lands with minimal water and fertilizer inputs. These traits make millets vital for ecosystem sustainability, crop diversification, and combating "hidden hunger", a condition caused by micronutrient deficiencies, that affects billions globally. Despite these advantages, millets, especially minor species such as little millet, have been neglected in genomic research, limiting their breeding potential.

This study fills a critical gap in little millet improvement by providing a near complete platinum-quality genome assembly. Combined with extensive transcriptome datasets[10,12] this work lays the foundation for advanced population genomics studies, enabling the exploration of genetic diversity and linking genetic variation to critical agronomic traits related to yield, environmental resilience, and micronutrient fortification. Our research reveals that the little millet genome is a

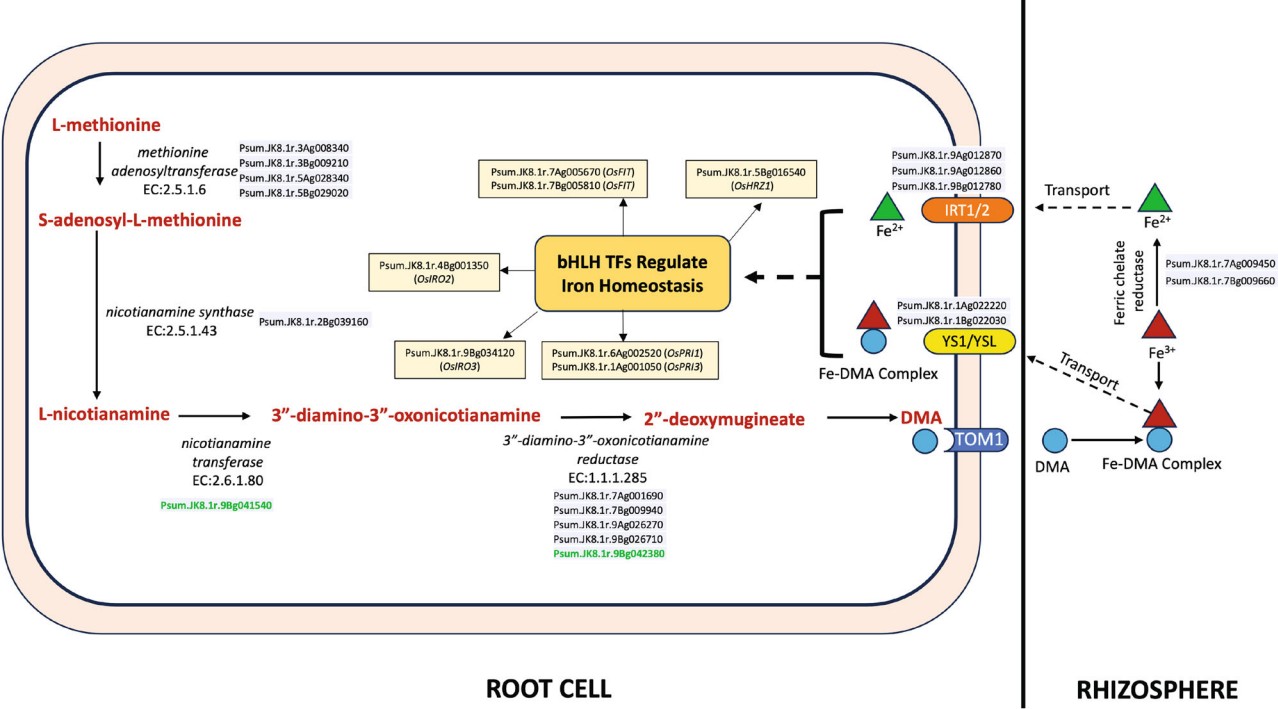

**Fig. 7 | Schematic representation of iron uptake and mobilization pathways in little millet.** The diagram showcases key genes from the JK-8 genome potentially involved in Fe reduction and Fe chelation pathways. Genes identified through LSV-viz analysis are highlighted in green.

**Table 3 | Micronutrient pathway genes in little millet identified via GWAS and LSV-viz analyses**

| Little millet gene | Analysis | Arabidopsis/rice ortholog | Micronutrient element |
|---|---|---|---|
| Psum.JK8.1r.2Ag029040 | LSV-viz | OsBMC | Fe, Mn, Cu |
| Psum.JK8.1r.2Bg030570 | LSV-viz | OsBMC | Fe, Mn, Cu |
| Psum.JK8.1r.3Ag000590 | LSV-viz | ZIP2 | Zn, Fe |
| Psum.JK8.1r.3Ag002970 | LSV-viz | MYB2; MYB2 transcription factor | Fe, Cd, Zn, Co, Mo |
| Psum.JK8.1r.3Ag002980 | LSV-viz | MYB2; MYB2 transcription factor | Fe, Cd, Zn, Co, Mo |
| Psum.JK8.1r.3Ag037860 | LSV-viz | PMA1, PMA2; H + -transporting ATPase | Na |
| Psum.JK8.1r.3Ag038080 | LSV-viz | SLC9A10_11; solute carrier family 9 (sodium/hydrogen exchanger), member 10/11 | Na |
| Psum.JK8.1r.9Bg041540 | LSV-viz | SPT; serine palmitoyltransferase | Fe |
| Psum.JK8.1r.9Bg042380 | LSV-viz | DMAS1; 3"-deamino-3"-oxonicotianamine reductase | Fe |
| Psum.JK8.1r.5Bg015260 | SNP-GWAS | PP3R, CNB; serine/threonine-protein phosphatase 2B regulatory subunit | K |
| Psum.JK8.1r.1Ag000230 | SNP-GWAS | AT2G21045 | As |
| Psum.JK8.1r.2Ag013440 | SNP-GWAS | CPS4; syn-copalyl-diphosphate synthase | Fe |

tetraploid, characterized by two sub-genomes that remain largely undifferentiated due to recent polyploidization. However, early signs of diploidization, including genome fractionation and sub-genome-specific expression bias that varies by gene and tissue, are evident. Polyploidy increases genetic diversity and expands the gene pool, enabling the retention of duplicate genes that can evolve new or improved functions. This genetic versatility likely contributes to little millet's remarkable adaptability to stress conditions and its resilience in marginal environments. Understanding sub-genome dominance offers new opportunities to target specific genes for improving key traits. Furthermore, its close relationship with broomcorn millet, which shares a similar evolutionary trajectory, enables the exchange of genetic and genomic knowledge and resources. This synergy between the two species can further accelerate advancements in breeding and crop improvement strategies.

The diversity panel comprising 300 accessions, along with the genomic and phenomic datasets generated in this study, provide a solid foundation for genetic interventions and developing marker-assisted selection strategies for crop improvement. The limited diversity within the scarce available germplasm of little millet, along with the frequent exchange of material among breeders, may have contributed to the lack of alignment between sub-populations and their geographic origins. Additionally, some advanced breeding lines have been derived from the same ancestral germplasm through hybridization and line development, resulting in genetic similarities across geographically distant regions. The repeated use of elite parental lines in breeding programs, combined with selection for specific agronomic traits, further reduced the geographic structuring of genetic variation. The identification of high-micronutrient accessions through population genetic analysis has immediate implications for breeding programs. The accessions which exhibit significantly higher micronutrient content, can be directly incorporated into breeding pipelines to develop nutritionally enriched varieties. Additionally, the identification of candidate genes for iron fortification opens the door

for advanced genetic interventions, including gene editing or transgenic approaches, to further optimize micronutrient concentration.

The identification of genotypes excelling in both micronutrient density and agronomic traits is particularly promising, as it addresses the common trade-offs between nutritional quality and yield[26]. Genotypes WV-260-1, WV-169, and GPMR-660, with their distinct trait combinations, offer a diverse genetic resource for developing multi-trait improved cultivars. Incorporating these genotypes into breeding programs can facilitate the pyramiding of favorable alleles for high yield and enhanced nutritional quality. Genotypes identified for high Fe and Zn levels can be strategically utilized in marker-assisted backcross programs to introgress alleles linked to these traits into commercial cultivars. Additionally, these genotypes can serve as founder lines in genomic selection programs, accelerating the development of new lines with superior traits through the enrichment of advantageous alleles. Insights from biofortification efforts in other grain cereals underscore the potential of leveraging diverse breeding materials to achieve both high grain yield and nutritional sustainability[6].

Through GWAS analyses, we identified loci associated with micronutrient accumulation, particularly for iron, a key nutrient for combating hidden hunger. Candidate genes involved in iron chelation, reduction pathways, and acquisition and transport into grains were identified, offering new targets for biofortification. Our findings provide insights on the molecular mechanisms underlying iron accumulation in little millet grains, highlighting genetic pathways that contribute to its nutritional superiority. The resolution of genetic loci influencing iron accumulation adds depth to the current understanding of biofortification strategies in minor millets. These discoveries also improve our understanding of how polyploid genomes can maintain both agronomic resilience and superior nutritional traits, providing a unique advantage over other staple crops. While this study identified promising loci and candidate genes, functional validation remains an essential next step to understand the molecular basis of Fe accumulation in little millet grains. The precise roles of genes involved in iron chelation and transport must be established through gene editing or overexpression studies. Additionally, while the GWAS leveraged genetic diversity within the studied population, expanding the analysis to include more diverse germplasm collections and environmental conditions could uncover additional loci and adaptive traits. Further efforts to improve traits like disease resistance, drought tolerance, and yield stability will enhance little millet's potential as a climate-resilient and nutritionally rich crop in resource poor and rainfed agroecological systems in Asia and Africa.

In summary, this study represents a significant step in elevating little millet from an underutilized orphan crop to a valuable agricultural resource. The high-quality genome assembly, combined with GWAS insights, provides a robust foundation for enhancing both agronomic performance and nutritional value, particularly in addressing micronutrient deficiencies. By bridging the knowledge gap in little millet genetics, this research unlocks its potential to contribute to global food security, improve nutrition, and support sustainable farming practices.

## Methods

### Plant material
The little millet cultivar JK-8, a widely cultivated variety across India, was used to generate the reference genome sequence for this species. JK-8 was released in 1987 as a cultivar by Jawaharlal Nehru Krishi Vishwavidyalaya, Jabalpur, Madhya Pradesh, India. Seeds of JK-8 were sourced from the All India Coordinated Research Project (AICRP) on small millets at the University of Agricultural Sciences, Bengaluru, India. In addition to JK-8, a total of 299 diverse little millet accessions, comprising breeding lines and germplasm collections from ten geographically distinct regions of India, namely, Andhra Pradesh, Bihar, Chhattisgarh, Gujarat, Karnataka, Madhya Pradesh, Maharashtra,

Odisha, Tamil Nadu, and Telangana (Supplementary Data 8), were assessed in this study (Supplementary Data 8). Seed material for these accessions was sourced from various institutions namely, ICAR-Indian Institute of Millets Research, Hyderabad; College of Agriculture, GKVK, University of Agricultural Sciences, Bangalore; University of Agricultural Sciences, Dharwad; Agriculture College, Dang, Gujarat; and S.G. College of Agriculture and Research Station, Jagdalpur, Chhattisgarh.

### DNA isolation, library construction and sequencing
Leaf tissue was collected from a three-week old plant of JK-8 grown under greenhouse conditions. The JK-8 plant selected for sequencing was derived through three generations of single-seed descent. High molecular weight DNA was extracted using CTAB method and quantified using the dsDNA Broad-Range assay on a Qubit 4 Fluorometer (ThermoFisher, USA). Quality control of the extracted DNA was conducted by determining the fragment length distribution analyzed on a Femto Pulse (Agilent, USA) pulsed field gel electrophoresis system. The DNA was fragmented using a Megaruptor 4 instrument (Diagenode, USA) and a PacBio SMRTbell library was constructed using the SMRTbell 3.0 library prep kit (PacBio, USA). HiFi reads were generated on a SMRT cell 25 M using the PacBio Revio system (PacBio, USA).

### Oxford nanopore sequencing
DNA was purified with Qiagen Genomic kit (Cat# 13343). DNA quality and concentration were tested using BioAnalyzer. About 4 micrograms of DNA sample was used for size selection of long DNA fragments using the PippinHT system (Sage Science, USA). End repair of DNA fragments and adapter ligation were performed according to the manufacturer's recommended protocol (Oxford Nanopore Technologies, UK) and the library was measured by Qubit 4.0 Fluorometer (Thermo Fisher Scientific, USA). The library was sequenced using Nanopore PromethION sequencer (Oxford Nanopore Technologies, UK).

### Hi-C library construction and sequencing
A Hi-C library was prepared from the JK-8 leaf tissue using the Proximo Hi-C (Plant) kit (Phase Genomics, USA). The library was quantified using the dsDNA High Sensitivity (HS) assay on a Qubit 4 Fluorometer (ThermoFisher, USA) and the average fragment length was assessed using the D1000 assay on a TapeStation 4200 system (Agilent, USA). Paired-end 150 bp reads were generated using Illumina NovaSeq 6000 platform.

### Genome assembly and scaffolding
PacBio HiFi reads, ONT data, and Hi-C data were provided to hifiasm (v0.19.9)[13] to generate the contig-level genome assembly. Subsequently, Hi-C data were mapped to the primary contig assembly using HiC-Pro (v3.1.0)[27], and scaffolding was carried out with the EndHiC pipeline (v1.0)[28]. The Hi-C contact map was generated using 3D-DNA (v180922)[29] script and visualized using Juicebox (v1.11.08)[30]. Manual corrections were performed to address misassembled regions. ONT reads were used for gap closing on the scaffold sequence using TGSGapcloser software[31] with the –racon parameter.

### Repeat annotation
The EDTA (v2.2.0) pipeline[32] was used for de novo annotation of repeats, including various classes of transposable elements (TEs) such as Long Terminal Repeats (LTRs) and DNA Transposons in both little millet and broomcorn millet independently. The raw repeat libraries generated by EDTA were manually curated and merged using the make_panTElib.pl script (included in the EDTA pipeline) with thresholds of 80% sequence identity and 80% coverage. Additionally, major tandem repeats were added to create a comprehensive draft repeat library. This finalized repeat library was used for genome-wide repeat masking with RepeatMasker (4.1.6)[33] to estimate the overall repeat

composition of the genome in both little millet and broomcorn millet. Full-length LTR retrotransposons with their insertion ages were extracted using LTR_retriever through EDTA pipeline[32], and TEsorter[34] was used to classify the LTRs into different families in both little millet and broomcorn millet independently. The LTR assembly index (LAI)[35] was used to assess the quality of the LTR assembly quality in the genome. Next, tandem repeats were annotated using Tandem Repeats Finder (TRF v4.09)[36], with additional repeats specific to the little millet genome incorporated into the final repeat library to ensure completeness.

For comparative repeat analysis between little millet and broomcorn millet, we applied a consistent annotation strategy to minimize methodological discrepancies and ensure reliable interpretation. The EDTA pipeline was run independently on each genome, and the resulting repeat libraries were merged using the PanTE module (PanEDTA) from the EDTA toolkit. To enhance completeness, tandem repeats identified by Tandem Repeat Finder (TRF) were added to the merged library. Both genomes were then masked using this comprehensive repeat library to estimate overall repeat content.

## Centromere and telomere localization based on repeat elements
Centromeric regions were identified by analyzing the distribution of centromere-specific tandem repeats (CentTR1 and CentTR2) and Gypsy-CRM LTR elements. This approach enabled the localization of putative centromeres and peri-centromeric regions across the assembled chromosomes. The distribution and copy number of telomeric repeats were identified based on the canonical 7-bp sequence (AAACCCT) within the top 100 kb of the short-and long-arm of the chromosome using Quartet (v1.2.2)[37] pipeline to determine the organization and completeness of telomeric regions across chromosomes.

## Gene annotation
Illumina RNASeq data from 10 tissue samples[12] covering three major growth phases: emergence [germinating seeds (GS), radicle (RD), plumule (PU)], vegetative [young leaf (YL), young root (YR), crown meristem (CM), vegetative stem (VS)], and reproductive [early panicle (PE), mid panicle (PM), late panicle (PL)], along with protein database from Broomcorn reference genome[14] were provided to Braker3[38–41] pipeline for annotating the protein-coding genes on the soft-masked genome. PASA (v.2.5.3) software was then used to further assemble and incorporate both little millet and broomcorn millet transcript alignment evidence into Braker gene annotation. The transcription factors (TFs) of JK-8 were identified using the PlantTFDB v5.0 prediction tool (http://planttfdb.gao-lab.org/prediction.php) and categorized into various families. The protein domain structures of all predicted protein-coding genes in little millet cv. JK-8 were determined using InterProScan version 5.72-103.0[42]. Functional annotation with KEGG and Gene Ontology (GO) terms was carried out using eggnog-mapper (version emapper-2.1.13)[43] leveraging eggNOG orthology data[44]. Sequence alignments were conducted using DIAMOND[45].

## Smudgeplot analysis
Smudgeplot was generated using k-mer frequencies in trimmed data to infer ploidy structure of the little millet genome[46].

## Sub-genome dominance analysis of homoeologous gene pairs
Previously published developmental transcriptome data[12] from ten distinct tissues, representing three major growth phases, emergence [germinating seeds (GS), radicle (RD), plumule (PU)], vegetative [young leaf (YL), young root (YR), crown meristem (CM), vegetative stem (VS)], and reproductive [early panicle (PE), mid panicle (PM), late panicle (PL)], were used for genome dominance analysis. All fully retained homoeologous gene pairs (19,147 pairs) were selected for sub-genome dominance analysis across ten tissues representing emergence, vegetative, and reproductive phases. Transcripts Per

Million (TPM) values were extracted from all RNA-seq samples, lowly expressed gene pairs were filtered out and the relative expression of each homoeolog was assessed. A gene pair was considered lowly expressed if the combined TPM of its A and B sub-genome copies was <1 in any RNA-seq sample. After filtering, relative expression levels of each homoeolog were calculated using normalized TPM values according to the following equations:

$$RE_A = \frac{TPM(A)}{TPM(A) + TPM(B)} \tag{1}$$

$$RE_B = \frac{TPM(B)}{TPM(A) + TPM(B)} \tag{2}$$

where RE represents the relative expression level, and A and B indicate the sub-genomes to which each homoeolog in the gene pair belongs. The final RE value for each developmental tissue was obtained by averaging RE values across all replicates, excluding gene pairs with low expression or significant outliers ($P$-value < 0.05) in any replicate. Dominance categories were defined based on the final RE of the B homoeolog ($RE_b$): A-dominant ($0 \leq RE_b \leq 0.2$), Balanced ($0.2 < RE_b < 0.8$), and B-dominant ($0.8 \leq RE_b \leq 1$). To assess whether expression dominance was stable or dynamic across the ten tissues or developmental stages, $RE_b$ values were compared to determine if each gene pair consistently fell within the same category. All $RE_b$ values were visualized using the R package ggplot2 (https://ggplot2.tidyverse.org).

## Synteny analysis
Synteny analysis was performed as described in ref. [47]. Briefly, sequence homology was assessed using BLASTP to compare the little millet proteins against the broomcorn millet proteome. BLAST hits with an E-value of 1e-20 or lower and within the top 40% drop from the highest bit score were retained for further analysis. Syntenic gene pairs between little millet and broomcorn millet were identified using DAGChainer[48] with default settings. The resulting collinear gene pairs were processed using a Perl script to filter for the highest chain-score, ensuring that each query sequence was represented only once in the final syntelog table. These filtered gene pairs were then mapped onto the little millet chromosomes to generate the syntelog table.

## PanKmer analysis of millet and related cereal species
To investigate the genetic relationships and shared genomic features between little millet, other millets, and related cereal species, we employed the PanKmer pipeline[49], a k-mer-based comparative genomics approach designed for identifying shared and unique sequences across multiple genomes. Two iterations were completed, one where chromosomes were compared between the two genomes, and one where the A and B genomes of little millet were compared to the A and B genomes of broomcorn.

High-quality genome assemblies for little millet, and other millet and related cereal species (e.g., broomcorn millet, foxtail millet, fonio millet, pearl millet, barnyard millet, finger millet, rice, maize, and sorghum) were obtained from public databases, including NCBI and Ensembl Plants. K-mers of length 31 (or a size optimized for the analysis) were generated for each genome using Jellyfish, a fast and memory-efficient tool for counting k-mers in large datasets. Default parameters were used unless otherwise specified. The PanKmer pipeline was run to compare the k-mer profiles across all species. The analysis was divided into two components: (1) Intra-species comparison: to identify shared and unique k-mers within the sub-genomes of polyploid species such as little millet and broomcorn millet, and (2) Inter-species comparison: to determine the extent of shared and unique k-mers between different species, highlighting conserved and divergent genomic regions. PanKmer-generated data were visualized as heatmaps to illustrate the overlap and uniqueness of k-mers among

the genomes. To assess genetic differentiation, pairwise similarity scores were calculated based on shared k-mer percentages. The relationships between sub-genomes within species (e.g., the two sub-genomes of little millet and broomcorn millet) and between species were analyzed to infer evolutionary divergence and polyploidy history.

To validate the PanKmer results, a phylogenetic analysis was performed using a data matrix consisting of 1295 common orthologs retained across little millet and other millet and cereal species. Sequences from individual orthologous gene sets were aligned with ClustalW v.2.1[50], and poorly aligned regions were trimmed using tri-mAL v.1.2[51]. The trimmed sequences were then concatenated using the Phyutility[52] program, resulting in a final data matrix with a total alignment length of 1,246,613 bp. Phylogenetic relationships were inferred using the maximum-likelihood method in RAxML v.8.2.12[53], employing rapid bootstrapping (100 replications) and the GTRGAMMA substitution model. The resulting phylogenetic tree was visualized using the interactive Tree of Life (iTOL) v.4 web server[54].

### Ks analysis

The distribution of synonymous substitutions (Ks) was performed as described previously[55]. In brief, for each pair of syntelogs between the sub-genomes of little millet or broomcorn millet, protein sequences were aligned using ClustalW v.2.1[50], and corresponding codon alignments were generated with PAL2NAL[56]. Ks values for each sequence pair were estimated using the maximum-likelihood method in codeml of the PAML package[57], applying the F3x4 model. All analyses were carried out using DATED, a Python-based wrapper that integrates the above steps, providing an efficient single-step solution for estimating the level of synonymous substitution (Ks) between paralogous and orthologous sequence pairs. Histograms were generated by log-transforming Ks values >0.001. Gaussian mixture models were fitted to the ln(Ks) values using the R package Mclust, and the number of Gaussian components, mean of each component, and data fractions were calculated. The mutation rate (λ) was estimated using the synonymous substitution rate (Ks = 0.57038) corresponding to the divergence between *Oryza sativa* and *Panicum miliaceum*, which took place approximately 41.4 to 51.8 million years ago[17], applying the following formula

$$\lambda = Ks/(2 \times \text{divergence time in My}) \qquad (3)$$

where λ represents the mutation rate and Ks is the rate of synonymous substitutions.

### The Bayesian method MCMCTree

A species tree was constructed using single-copy orthologs from 14 plant species with IQ-TREE[58] to infer evolutionary relationships. Divergence times were estimated using MCMCtree with an independent rates model and nucleotide sequence data. Calibration points were based on TimeTree[17]: *Sorghum bicolor*–*Zea mays* (9–11.8 mya), *Oryza sativa*–*Panicum miliaceum* (41.4–51.9 mya), and *Arabidopsis thaliana*–*Medicago truncatula* (102–112.5 mya). The resulting tree was visualized using FigTree (https://tree.bio.ed.ac.uk/software/figtree/).

### Visualization

Collinearity detection was performed by MCScanX[59] through Blast alignment results between the two sub genomes. To remove overlapping duplicated regions and refine the final visualization, the blast results were filtered against the syntelog table (Supplementary Data 4) that restricted the matches to their first accurate hit. The resultant collinear blocks were then visualized using SynVisio[60]. Additionally, gene density and repeat density tracks were generated in windows of 1 million base pairs and added as histogram and heatmap tracks respectively. Finally, gene expression values were also generated using the developmental transcriptome data[12] by aggregating TPM

(transcripts per million) values in windows of 1 million base pairs and added as a heatmap track.

### Resequencing of diversity panel

A panel of 300 little millet accessions was sequenced at 5–7× coverage using Illumina short read sequencing technology. Plants of each accession were grown in 3-inch pots and one leaf was harvested from 3-week-old plants. DNA was extracted using sodium dodecyl sulfate (SDS) DNA extraction method. Library preparation was carried out using the standard method of Twist library preparation enzymatic fragmentation (EF) kit 2.0 (Twist Bioscience, USA). Sequencing was performed on illumina Novaseq X plus platform.

Raw reads were trimmed with cutadapt[61] and quality assurance was performed with FastQc[62] on both the raw reads and the trimmed reads. The trimmed reads were aligned to the JK-8 reference genome using bwa[63] and alignment statistics were compiled using samtools[64] and bamtools[65]. Variants were identified using bcftools mpileup and called using bcftools call[66]. The resulting per-sample variant files were filtered using quality score 30, and minimum depth 5 before merging and then filtered again using max missing 0.8, and max maf 0.05 using vcftools[67].

### Genetic relationships and linkage disequilibrium

The SNPs identified in the diversity panel were filtered for quality score of 30, maximum missing values of 15%, maximum heterozygotic alleles of 15%, and minimum allele frequency of 0.05. A total of 249,511 SNPs across the 18 chromosomes were used for genetic diversity analysis and association mapping.

The population structure of the diversity panel was assessed upon estimating the most likely number of clusters (K), and the degree of admixture between the clusters, using the program fastSTRUCTURE[68]. To determine the optimal number of groups (K) and assign group membership for each genotype, we employed the "chooseK.py" function in fastSTRUCTURE, testing values of K ranging from 1 to 20, as described by ref. 68. The value of K that best fits the most likely number of clusters was determined based on the lowest prediction error, and the smallest number of iterations for convergence.

In addition to fastSTRUCTURE, population structure was assessed using ADMIXTURE, STRUCTURE and discriminant analysis of principal components (DAPC). ADMIXTURE v1.3.0[69] was run for K = 1 to 25 and the optimal K was determined based on the lowest fivefold cross-validation (CV) error. STRUCTURE[70] was performed for K = 1 to 15 with 50,000 burn-in and 50,000 MCMC iterations per run and the best K was inferred using DK method[71]. DAPC was performed using the "adegenet" package in R[72], with clusters identified through Bayesian information criterion (BIC)-optimized K-means clustering and 200 principal components retained based on a-score optimization to maximize discrimination.

Principal component analysis was performed using the "adegenet" package in R. The genetic distance between populations was calculated using Nei's distance method. The analysis was conducted using the "adegenet" (version 2.1.10) and "poppr" (version 2.9.6) R packages. The resulting genetic distances were visualized using the "dendextend" R package (version 1.19.0).

LD decay was calculated using "popLDdecay" (version 3.43). The LD decay curve was fitted to the scatterplot using smoothing spline regression, following the procedure outlined by ref. 73. The point of intersection between the LD curve and the predefined r² threshold was determined to define the LD decay distance.

### Visualization of large structural variants (LSV-viz)

Using the BAM files generated during SNP analysis, bedtools coverage[74] was applied to determine the gene coverage for all 300 samples. Gene counts for each sample were compiled into a data matrix, TPM normalized per sample, and then used as input for

**LSV-viz.** LSV-viz was developed in python using NumPy, Pandas, and matplotlib modules and Python 3. The reference sample (JK-8) served as the denominator when calculating the gene coverage ratio for all samples. These ratios were visualized using a sliding window average of 20 and a step size of 5. LSVs were then identified, and their corresponding gene and genome coordinates were extracted for further analysis.

Genes identified within the large structural variants (LSVs) from LSV-viz were compared to a known set of previously identified ionome genes[23]. Lines from the diversity panel containing LSVs with genes overlapping the ionome genes were identified. These lines were then compared to outliers in the phenotype dataset using the IQR method for outlier detection. Lines with ionome genes in their LSVs that also exhibited outlier phenotypes were reported.

### Structural variant characterization and analysis

Structural variants were identified using two complementary tools, Delly (v1.1.6) and Manta[75], both applied to BAM files generated during SNP analysis. SV calling was performed independently for each sample with both tools. Raw SVs from each caller were filtered using BCFtools (v1.8), retaining only those with a PASS flag and a minimum read depth of 2. Variants smaller than 30 bp or larger than 2 Mb, as well as translocations, were excluded due to challenges in accurate validation with short-read data.

The filtered SVs from all samples were merged across individuals using the SURVIVOR tool (https://github.com/fritzsedlazeck/SURVIVOR). We compared the SVs identified by both callers and distinguished shared (consensus) variants (14,273 SVs) from those uniquely detected by either tool. To assess the reliability of the SV calls, we performed read-depth-based validation using samplot[76], which confirmed high support for shared SVs and provided strong evidence for many uniquely called variants as well. Given the concordance, consensus SV sets were used for downstream analyses, including GWAS.

### Field phenotyping of the diversity panel

The diversity panel of 300 accessions was evaluated for agronomic traits and grain mineral content in a replicated field experiment at two locations. The field experiment was conducted during the *Kharif season of* 2024 at Gandhi Krishi Vigyana Kendra (GKVK), Bengaluru, Karnataka, India (13.0801° N latitude, 77.5785° E longitude, altitude 930 m above MSL) and ICAR-Indian Institute of Millets Research (IIMR), Hyderabad, India (17.3205° N latitude, 78.3961° E longitude, altitude 542 m above MSL). The experiment was designed using alpha lattice design with three replications. Each accession was planted in a single 3-meter row (plot), with 22.5 cm spacing between rows and approximately 10 cm between plants resulting in about 30 plants per accession per replicate. Seedlings were thinned at 15 days after emergence to achieve optimum plant population in each accession. Standard crop management practices were followed throughout the growing season to ensure healthy growth of the little millet accessions. Observations on days to 50% flowering (DFF), thousand seed weight (TW) and grain yield per plant (GYPP) were recorded according to the standard operating procedure (SOP) for small millets. DFF was recorded on a plot basis, while TW and GYPP were recorded on five representative plants per plot.

### Quantification of seed mineral elements

Panicles of three plants in each plot were covered with parchment paper bags at the time of emergence to ensure self-pollination and maintain the genetic purity of the seeds. After harvesting the panicles at physiological maturity using sterilized scissors, they were placed in parchment paper bags and dried on clean tarpaulin sheets to prevent contamination with soil, dust, or any metal. The panicles were then hand-threshed on plastic sheets using a clean wooden bar, and gloves

were worn during this process. The seed samples were washed with distilled water to remove any impurities, oven-dried at 40 °C for 24 h, and then stored in sealed plastic or paper bags at ambient temperature to preserve the quality of the seeds.

Seven mineral elements namely, Ca, Fe, K, Mg, Na, P, and Zn were quantified using an Inductively Coupled Plasma–Optical Emission Spectrometer (ICP-OES) (Agilent Technologies 5800 Series, USA). For reliable and efficient elemental analysis, a multi-standard solution containing known concentrations of multiple elements was used to calibrate the ICP-OES instrument. All the glassware was soaked in a dilute nitric acid solution, followed by thorough rinsing with distilled water, and the chemicals used in the analysis were of analytical grade. To prepare a 100 ppm of a multi-element standard solution, 2.5 mL of the stock solution (1000 μg/mL) was transferred into a 250 mL volumetric flask and diluted to volume with Milli-Q water. For the calibration graph, standard solutions with final concentrations ranging from 0.2 ppm to 25 ppm were used. Approximately 0.5 to 1 gram of each test sample was weighed into a clean silica crucible, pre-ignited in a hot plate for 10–15 min at 100 °C. The test samples were then transferred to a furnace at 650 °C for 2 h for the ashing process and cooled to room temperature in a desiccator. The ash was dissolved in 5 ml of 10% aquaregia (0.125 mL of conc. $HNO_3$@69%; 0.375 mL of concentrated HCl@36%) upon boiling and diluted to 25 mL with Milli-Q water in a volumetric flask.

The concentration of each element in the sample was calculated using the formula:

$$C = \frac{R \times V \times D.F.}{S} \tag{4}$$

where C = conc. of the element in the sample, mg/kg; R = reading in ICP-OES, mg/L; V = volume used, mL; S = sample weight used; D.F. = dilution factor.

### Ionome analysis

A comprehensive list of functionally characterized gene sequences of rice and Arabidopsis[23] were retrieved from The Rice Genome Annotation Project Database (https://rice.uga.edu) and The Arabidopsis Information Resource (https://www.arabidopsis.org), respectively. These protein sequences were clustered based on protein-protein sequence homology analysis, and little millet orthologs were identified using Orthofinder (https://github.com/davidemms/OrthoFinder). A curated list of ionome-related genes[23] was used to identify the orthologs in little millet.

### Genome-wide association mapping for agronomic traits and grain mineral composition

Association mapping for three agronomic traits (DFF, TW, and GYPP), and grain micronutrient composition (Ca, Fe, K, Mg, Na, P, and Zn) was determined using Genome Association and Prediction Integrated Tool (GAPIT)[77] based on 249,511 SNPs and phenotypic data from both test locations and best linear unbiased predictions (BLUPs) calculated based on pooled data. The Q values, which considered the genetic structure of the diversity panel, PCA and the kinship coefficient matrix (K) were used in the analysis. Two models, namely MLMM and BLINK[78] models were used for association analysis. The quantile-quantile (Q-Q) plots of an individual trait were compared between the models to declare a significant marker-trait association.

Haplotype analysis for significant marker-trait associations (MTAs) was performed using the "*geneHapR*" R package (version 1.0.4) to dissect allelic variation and evaluate haplotype-phenotype relationships[79]. A ± 1 Mb genomic region surrounding each significant SNP was extracted to define haplotype blocks. SNPs with missing and heterozygous genotype calls were removed to ensure data quality. Only haplotype groups comprising at least three genotypes were

retained for downstream statistical analysis[80]. Phenotypic differences among haplotype groups were assessed using one-way analysis of variance (ANOVA), followed by Tukey's Honest Significant Difference (HSD) post hoc test to determine statistically significant pairwise differences in trait expression.

Association mapping based on structural variants identified among the diversity panel was conducted using the rMVP packages[81] in R statistical software[82]. cMLM and BLINK models were used for association analysis. The Bonferroni correction method was applied to determine the significance of marker-phenotype associations. LD triangle plot for significant SNPs on chromosome 3 associated with Fe was visualized using "LDBlockShow" (Version 1.40).

### Statistical analysis
Outliers in the data were detected using the Z-score test and then removed. Adjusted means of replicates for agronomic and mineral traits were obtained by fitting mixed linear models (MLM). These adjusted means were calculated as the best linear unbiased predictions (BLUPs), considering replication, blocks and genotypes as random effects, while location was treated as a fixed effect. This analysis was performed using the "lme4" package in R[83]. In this model, Yik represents the trait of interest for the kth genotype in the ith replicate and the jth location. The term μ denotes the overall mean effect. Lj captures the fixed effect of the jth location, while Ri(j) represents the random effect of the ith replicate nested within the jth location. Additionally, Bl(i,j) accounts for the random effect of the lth block nested within the ith replicate and jth location. Gk reflects the random effect of the kth genotype. The interaction term (G×L)kj represents the genotype-by-location interaction. Finally, €ik is the residual error term, which is assumed to follow a normal distribution with a mean of zero and constant variance $\sigma2$.

Spearman correlation coefficients for the agronomic and nutritional were calculated to assess the linear relationship between the variables. The analysis was performed using the "scipy.stats" module in Python (version 3.11.0) and the statistical significance of the correlations was evaluated at a significance level of $p < 0.05$ and 0.01. Correlation coefficients were visualized using "matplotlib" module (version 3.10.0) to generate scatter plots with upper triangular matrix displaying the correlation coefficients, and a lower matrix with dot plots and best-fit lines.

### Reporting summary
Further information on research design is available in the Nature Portfolio Reporting Summary linked to this article.

## Data availability
Raw sequencing data have been deposited in National Center for Biotechnology Information (NCBI) under the BioProject accession PRJNA1204366. The genome assembly and annotation files are available at Zenodo [https://zenodo.org/records/16776976]. The Bio sample IDs for raw sequencing data from the diversity panel are listed in Supplementary Data 21. Source data are provided with this paper.

## Code availability
Code for SynViso (custom genome visualizer) and DATED (*Ks* analysis) have been deposited in the public domain SynVisio [https://github.com/kiranbandi/synvisio] and DATED. Custom python, R and shell scripts that were used to assemble and annotate genome, perform genome dominance analysis, and population genetics are provided in Zenodo [https://zenodo.org/records/17524940].

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

## Acknowledgements
This work was co-funded by the Global Centre of Excellence on Millets (Shree Anna) project, ICAR-Indian Institute of Millets Research, Hyderabad, India (N.T.); Gandhi Krishi Vigyana Kendra (GKVK), University of Agricultural Sciences, Bangalore, India (MKP); and Agriculture and Agri-Food Canada (RS). M.W.K. and S.K. were supported by the National Research Council Canada through the strategic initiatives of the Aquatic and Crop Resource Development Research Centre. K.K.G. and L.K.D. were supported by the Crop Development Centre, University of Saskatchewan. The authors extend their gratitude to Shweta Kalve for technical assistance and to Dustin Cram for support with data management. The authors also thank Dr. Boraiah B. for providing access to land and farm equipment for field experiments at GKVK.

## Author contributions
S.K., K.K.G., N.T., M.K.P. and R.S. conceptualized the study. R.S. and S.P. conducted PacBio HiFi sequencing, while N.T., M.K.P. and K.B.P. carried out ONT sequencing. Genome assembly, along with the annotation of repeats and protein-coding genes, was performed by K.C.K., S.P. and S.K. Genome dominance analysis was performed by P.G. Phylogenetic and Ks analyses were undertaken by S.K. The diversity panel was assembled by K.K.G., N.T., K.B.P., M.K.P. and H.E.P., with sequencing of the panel performed by K.K.G., M.K.P., K.B.P., N.T. and S.K. SNP analysis was conducted by M.W.K., S.K. and K.K.G. SV and SV-based GWAS analyses were handled by S.P., N.T., R.C. and A.G.S., while SNP-based diversity and GWAS analyses were performed by K.K.G., N.T., S.N., T.C.S., D.B., H.E.P., L.K.D., V.B.R.L., S.T.V., R.M. and T.D.W. Field experiments and elemental analyses were carried out by G.K.N., K.B.P., N.T. and M.K.P. T.D.W. and T.C.S., provided critical inputs in designing the experiments and editing the manuscript. V.B. contributed to data visualization. H.B.M. conducted the pathway analysis. R.C. performed MCMCtree analysis. The manuscript was written by K.K.G., S.K. and L.K.D., with input and contributions from all authors to the final version.

## Competing interests
The authors declare no competing interests.
