## [Peer Review file · Nature Communications]

Little millet genome reveals evolutionary insights into tetraploid structure and genetic basis of micronutrient density

Corresponding Author: Dr Sateesh Kagale

Version 0:

Reviewer comments:

Reviewer #1

(Remarks to the Author)

Gali et al. presented the first chromosome-scale genome assembly of little millet, elucidating its recent tetraploid structure, sub-genome dominance, and the genetic basis for micronutrient content. Through sequencing a diversity panel of 300 accessions and identifying key loci, the study provides valuable genomic insights for molecular breeding to enhance food security and sustainability. However, several concerns need to be addressed, and further analyses should be performed to fully explore the generated data before this manuscript can be considered for publication in Nature Communications.

1. While the JK-8 genome assembly is of high quality, 19 gaps remain. The authors should clarify why existing PacBio HiFi and ONT data were not utilized to close these gaps, as mature bioinformatics tools such as TGS-GapCloser and quarTeT are available. Supplementary Data 3 shows centromere sizes varying from 300 kb to 3100 kb. It is unclear whether the correct standard was used for centromere classification, and the methods for centromere detection are insufficiently described. Similarly, Supplementary Data 4 reveals missing telomeres on some chromosomes. The authors should specify the criteria (e.g., number and length of (AAACCCT) repeats) used to define telomere regions. Closing the assembly gaps would likely improve centromere and telomere detection, enhancing genome completeness for future use.

2. The study lacks Smudgeplot analysis, which can provide evidence of polyploid status. Additionally, the RNA-seq data used for annotation were previously published, but details about the tissues analyzed are absent. This information is crucial for assessing the sufficiency of the data.

3. The authors briefly mention higher expression in sub-genome A compared to sub-genome B across 10 tissues. More comprehensive analysis, such as allele-specific expression in different tissues, would provide deeper insights into sub-genome-specific functionality.

4. The methodology for comparative repeat analysis between little millet and broomcorn millet is unclear. Did the authors analyze both species using the EDTA pipeline, or were results for broomcorn millet derived from prior studies? Differences in methodology could lead to conflicting interpretations.

5. Estimation of divergence time using Ks values assumes a consistent mutation rate (μ), which may vary across periods and genes. Correcting Ks values for downstream analysis, as demonstrated in studies such as <https://doi.org/10.1104/pp.16.01981> and <https://doi.org/10.1093/molbev/msx242>, is recommended. Bayesian methods like MCMCTree would provide more robust divergence estimates and could reveal differing evolutionary histories among homologs.

6. In line 383, the assertion that adjusted P-values are "too stringent" lacks justification. Reducing the threshold to capture "true" associations risks introducing false positives. Experimental validation is needed to substantiate GWAS results. Furthermore, only one tool was used to detect SVs, despite the limitations of short-read sequencing, which may increase false positives. Employing multiple tools would improve reliability.

7. For high copy numbers of ionome-related genes, mapping depth should be checked to rule out assembly errors. A comprehensive gene family analysis, including distribution patterns, expression levels, selective pressures, and duplication origins, would better elucidate their roles. Haplotype analysis and genotype-phenotype associations for candidate genes are also recommended.

8. The lack of alignment between sub-populations and geographic origins, mentioned in line 280, merits further investigation. Possible causes such as introgression or incomplete lineage sorting (ILS) should be explored. Additionally, population analysis based on SV markers could provide new insights.

Minor Concerns:

- 1.The relevance of Figure 2C in illustrating translocations between Chr4 and Chr5 is unclear.
- 2.Did the authors evaluate collapsed regions when reporting high short-read mapping rates?
- 3.Why does the manuscript emphasize PFAM annotations in lines 178–184? Other databases like GO and KEGG should be considered, as these results are absent from Supplementary Data 2.
- 4.Line 208 needs a reference or evidence for the claim about gene expression influencing duplicated gene fate post-WGD.
- 5.Line 258 should include evidence for Gypsy-Ogre elements' role in genome stability and evolutionary pressures affecting repeat content (line 261).
- 6.Supplementary Fig. 7 does not match the manuscript; clarify chromosome 12's role and the meaning of LD lengths of 0.6 Mb and 8.1 Mb.
- 7.Parameters for merging SVs should be detailed.
- 8.Figure 4A lacks clarity on the meaning of label and branch colors.
- 9.A complete code repository is essential for reproducibility. Github Links to tool pages are insufficient.

Reviewer #2

(Remarks to the Author)

The manuscript that entitled chromosome scale assembly of little millet reveals evolutionary insights into tetraploid genome structure and genetic basis of micronutrient density, reported the release of a high quality chromosome-scale genome sequence of little millet elite cultivar JK-8, which is the first one in this species. Deep analyses of the sequence confirmed the tetraploid nature of little millet genome that is very similar to the genome of broomcorn millet. GWAS analysis with 300 diverse accessions of small millet collected from different growing regions of India identified genomic regions associated with high micronutrient contents. Those data combined with other genome comparisons with other cereals provides useful foundation for not only little millet improvement but also benefits other cereals future breeding.

In line 279, the authors analyzed the population structure of little millet accession using fastStructure and classified 300 accessions into 10 sub-population based on the fact that the marginal likelihood plateauing at $K = 10$. It is stated that the sub-population did not align with their geographic origins. The authors could test ADMIXTURE and STRUCTURE if these tools can help with a robust estimation of population structure. Also, DAPC is a tool to determine the optimal K . An accurate estimation of population structure is essential for the GWAS analysis. Therefore, these analyses may help the authors to refined their GWAS results.

Version 1:

Reviewer comments:

Reviewer #1

(Remarks to the Author)

The authors have made commendable efforts to conduct additional analyses and substantively enrich the manuscript. However, one prior concern remains: a complete, centralized code repository is essential for reproducibility. Links to tool homepages are insufficient. Please upload all code and scripts necessary to reproduce each stage of the study (e.g., genome assembly, genome annotation, gap filling, and population analyses), rather than only a subset of scripts. Once this is addressed, I will be happy to recommend acceptance of the manuscript.

Reviewer #2

(Remarks to the Author)

The authors performed additional analyses on population structure using ADMIXTURE and STRUCTURE. DAPC was also used to determine the number of clusters. I am glad that these analyses helped the authors to identify consistent GWAS results across different parameter settings. Overall, the manuscript has been significantly improved in this revision. I have no further questions.

Responses to reviewer comments

Reviewer #1:

Gali et al. presented the first chromosome-scale genome assembly of little millet, elucidating its recent tetraploid structure, sub-genome dominance, and the genetic basis for micronutrient content. Through sequencing a diversity panel of 300 accessions and identifying key loci, the study provides valuable genomic insights for molecular breeding to enhance food security and sustainability. However, several concerns need to be addressed, and further analyses should be performed to fully explore the generated data before this manuscript can be considered for publication in Nature Communications.

Response: We thank the reviewer for the positive evaluation of our study and for recognizing the significance of our work on the little millet genome and its potential contributions to food security and crop improvement. We have carefully addressed all the concerns raised, and detailed responses to each comment are provided below. Corresponding revisions have been made to the manuscript as outlined in our responses. We have provided both a tracked-changes version (marked) and a clean copy of the manuscript.

1. While the JK-8 genome assembly is of high quality, 19 gaps remain. The authors should clarify why existing PacBio HiFi and ONT data were not utilized to close these gaps, as mature bioinformatics tools such as TGS-GapCloser and quarTeT are available.

Response: We appreciate the reviewer's suggestion. In response, we ran the TGS-GapCloser software using available ONT and PacBio reads. This effort successfully closed 8 out of the 19 remaining gaps, specifically in chromosomes 4A, 4B, 5B, 6A, 7A and 9B. Some of the gaps filled were relatively small, ranging from approximately 200 bp to 1 Kb. The gap filling created a single contig chromosomes for Chr4B, Chr6A, Chr7A and Chr9B. These improvements have been incorporated into a revised assembly, now released as version 2.0. Corresponding updates have been made to both the Methods (lines 597-598, marked copy) and Results (lines 133 -141) sections of the manuscript to reflect this work.

Supplementary Data 3 shows centromere sizes varying from 300 kb to 3100 kb. It is unclear whether the correct standard was used for centromere classification, and the methods for centromere detection are insufficiently described.

Response: The centromere sizes reported in Supplementary Data 3 are putative and include both the core centromeric regions and surrounding pericentromeric regions, as they were inferred based on the distribution of centromere-associated tandem repeats (CentTR1 and CentTR2; Supplementary Figure 3) and centromeric retrotransposons (CRMs). We observed a high abundance of these centromeric repeats across the centromeres of most chromosomes, with notably fewer detected on Chr1A, Chr4A, and Chr9B, which may reflect assembly gaps or differences in chromosomal architecture. We have included a detailed description of the centromere identification procedure in the Methods (lines 572-574) section to clarify our approach and updated the text in Results section (lines 143-148).

Similarly, Supplementary Data 4 reveals missing telomeres on some chromosomes. The authors should specify the criteria (e.g., number and length of (AAACCCT) repeats) used to define telomere

regions. Closing the assembly gaps would likely improve centromere and telomere detection, enhancing genome completeness for future use.

Response: As described in the Methods (lines 724-728) section, telomeric regions were identified based on the presence of the canonical 7-bp telomeric repeat (AAACCCT) within 100 kb of the chromosomal ends (both short and long arms). To support this analysis, we used quarTeT to quantify telomeric repeat sequencing coverage. The number of telomeric repeats found in each chromosome are provided in Supplementary Data 4. While we acknowledge that unresolved assembly gaps likely contribute to the absence of detectable telomeric sequences on some chromosome termini, our gap-closing efforts using the available ONT data did not recover the missing telomeres. We agree that further gap closure, particularly using ultra-long reads, will be essential to improve the resolution of both centromeres and telomeres. We have noted this limitation in the Results (lines 149-154) section of the manuscript.

2. The study lacks Smudgeplot analysis, which can provide evidence of polyploid status.

Response: A Smudgeplot analysis was performed, which confirmed that the little millet genome is polyploid. The results revealed patterns indicative of allotetraploidy. We have included a description of this analysis in the Results (Line 210) and Methods (Line 743) sections, and the corresponding Smudgeplot has been added to Figure 2 (Fig. 2B).

Additionally, the RNA-seq data used for annotation were previously published, but details about the tissues analyzed are absent. This information is crucial for assessing the sufficiency of the data.

Response: We thank the reviewer for identifying this oversight. We have now added detailed descriptions of the tissues used in the RNA-seq analysis to the manuscript (Lines 172-174 and Lines 729-732). The little millet (genotype JK-8) developmental transcriptome was constructed using RNA-seq data from ten tissue types spanning three major growth phases: emergence [germinating seeds (GS), radicle (RD), plumule (PU)], vegetative [young leaf (YL), young root (YR), crown meristem (CM), vegetative stem (VS)], and reproductive [early panicle (PE), mid panicle (PM), late panicle (PL)]. These samples broadly capture gene expression across the life cycle of little millet, from germination through maturity.

3. The authors briefly mention higher expression in sub-genome A compared to sub-genome B across 10 tissues. More comprehensive analysis, such as allele-specific expression in different tissues, would provide deeper insights into sub-genome-specific functionality.

Response: To address this concern, we conducted a genome-wide assessment of sub-genome expression dominance using RNA-seq data from ten distinct tissues spanning the emergence, vegetative, and reproductive stages (Fig. 2D). This analysis revealed that only 3.5–9.7% of expressed homoeologous pairs were A-sub-genome dominant, while 3.4–9.8% were B-dominant across multiple tissues. Additionally, we identified dynamic gene pairs, those whose dominance varied across tissues, and stable gene pairs that maintained consistent dominance. Of the expressed homoeologous pairs, 79.0% were stable and 21.0% were dynamic (Supplementary Figure 6). These findings indicate that sub-genome expression bias in little millet is largely gene- and tissue-specific, rather than reflective of a consistent genome-wide dominance pattern. We have updated the Results (lines 235-246) section to reflect this revised analysis, and a methods (lines 745-769) sections describing this analysis has also been added.

4. The methodology for comparative repeat analysis between little millet and broomcorn millet is unclear. Did the authors analyze both species using the EDTA pipeline, or were results for broomcorn millet derived from prior studies? Differences in methodology could lead to conflicting interpretations.

Response: To avoid potential discrepancies in methodology that could lead to conflicting interpretations, we applied the same repeat annotation approach to both genomes. Specifically, we ran the EDTA pipeline independently for each genome, followed by merging the repeat libraries using the PanTE library construction module (PanEDTA) from the EDTA toolkit. To ensure completeness, we further updated the merged library with tandem repeats identified through Tandem Repeat Finder (TRF). Finally, both genomes were masked using this consolidated repeat library to estimate the overall repeat content. Detailed steps outlining this methodology are provided in the Methods section (lines 714-720).

5. Estimation of divergence time using Ks values assumes a consistent mutation rate (μ), which may vary across periods and genes. Correcting Ks values for downstream analysis, as demonstrated in studies such as <https://doi.org/10.1104/pp.16.01981> and <https://doi.org/10.1093/molbev/msx242>, is recommended. Bayesian methods like MCMCTree would provide more robust divergence estimates and could reveal differing evolutionary histories among homologs.

Response: Thank you for this insightful and important suggestion. In response, we have implemented the Bayesian method MCMCTree to estimate divergence times across species, as it provides a more robust evolutionary framework and accounts for rate variation across lineages. The Results section has been updated accordingly, and Figure 3B has been revised to reflect these new divergence estimates. Additionally, a new subsection detailing the MCMCTree analysis has been added to the Methods section (lines 826-832).

For consistency, we also estimated the mutation rate using the synonymous substitution rate ($K_s = 0.57038$) associated with the divergence between *Oryza sativa* and *Panicum miliaceum*, which occurred approximately 41.4–51.8 million years ago (Kumar et al., 2022). This divergence event was used as a calibration point in the MCMCTree analysis. Consequently, the divergence times between the progenitor species of little millet and broomcorn millet, as well as their sub-genomes, have been revised based on this integrative approach. Figure 3E and its accompanying text in the Results (lines 260-272) and Methods (lines 821-832) sections have been updated.

6. In line 383, the assertion that adjusted P-values are "too stringent" lacks justification. Reducing the threshold to capture "true" associations risks introducing false positives. Experimental validation is needed to substantiate GWAS results.

Response: We sincerely thank the reviewer for bringing this important point to our attention. We analyzed 249,511 SNPs using a mixed linear model (MLM). Applying a Bonferroni correction to control the family-wise error rate leads to a highly stringent significance threshold ($P = 0.05 / 249,511 \approx 2.0 \times 10^{-7}$), which corresponds to $-\log_{10}(P) \approx 6.7$. Such a strict threshold is statistically appropriate but may be overly conservative in the context of complex, quantitative traits controlled by multiple minor-effect loci. Hence, to capture potential true associations that were missed by Bonferroni correction, we followed a similar method used by Deng et al. (2021).

We dropped the statement “too stringent” in the revised manuscript and provided an explanation that to capture the potential true associations contributed by multiple minor effect loci below Bonferroni threshold, we followed a similar method used by Deng et al. (2021). (lines 468-471).

Furthermore, only one tool was used to detect SVs, despite the limitations of short-read sequencing, which may increase false positives. Employing multiple tools would improve reliability.

Response: Thank you for this valuable suggestion. We have now incorporated an additional SV caller, Manta (Chen et al., 2016), alongside our original pipeline based on Delly. Structural variant (SV) calling was performed independently using Delly and Manta, and the resulting SV sets were merged using the SURVIVOR tool. This approach yielded a high-confidence set of 14,273 consensus SVs detected by both callers. To assess reliability, we performed read-depth-based validation using Samplot, which confirmed that consensus SVs had significantly higher support compared to tool-specific variants. Based on this validation, only the consensus SVs (14,723) were retained for downstream GWAS analyses, thereby enhancing the robustness and comprehensiveness of our findings. The manuscript has been updated accordingly (Please check revised Results and Methods section; lines 352-375 and lines 902-906).

7. For high copy numbers of ionome-related genes, mapping depth should be checked to rule out assembly errors. A comprehensive gene family analysis, including distribution patterns, expression levels, selective pressures, and duplication origins, would better elucidate their roles. Haplotype analysis and genotype-phenotype associations for candidate genes are also recommended.

Response: We examined the mapping depth of high-copy ionome-related genes to assess the reliability of their copy number estimates and to rule out potential assembly artifacts. Our analysis confirmed consistent and elevated read depth across these loci, supporting their high-copy status, and rule out any assembly errors. A comprehensive analysis of gene family characteristics, such as distribution patterns, expression levels, selective pressures, and duplication origins is currently the focus of an ongoing student’s project in our group and will be published separately in the coming months.

Regarding the suggestion on haplotype analysis, we conducted haplotype analysis for the five significant marker-trait associations described in the results section to dissect allelic variation and evaluate haplotype-phenotype relationships (Zhang et al., 2023). The methods (lines 988 -996) and results section (line 500) of the manuscript have been updated. Supplementary Figure 19 has been added to present the results of haplotype analysis.

8. The lack of alignment between sub-populations and geographic origins, mentioned in line 280, merits further investigation. Possible causes such as introgression or incomplete lineage sorting (ILS) should be explored. Additionally, population analysis based on SV markers could provide new insights.

Response: In little millet, the overall diversity of available germplasm is relatively limited, and breeding efforts often rely on a common pool of elite lines. The observed lack of alignment between sub-populations and geographic origin likely reflects extensive germplasm exchange among breeders across different regions of India. Many of these lines are derived from shared parental material

through hybridization and selection. Consequently, each sub-population comprises accessions from multiple geographic regions. This pattern indicates that processes such as introgression and a shared breeding history, rather than strict geographic isolation, are key factors shaping the current population structure. Moreover, strong selection for a narrow set of agronomic traits (e.g., yield, maturity, stress tolerance) may have further homogenized the genetic backgrounds, thereby weakening any geographic differentiation.

Minor Concerns:

1. The relevance of Figure 2C in illustrating translocations between Chr4 and Chr5 is unclear.

Response: Hi-C contact maps show continuous interaction signals across the translocation breakpoint, supporting the validity of the translocation and indicating it is not an assembly artifact. However, to avoid redundancy and potential confusion, since the same data are duplicated in Supplementary Figure 2, we have removed Figure 2C from the main text.

2. Did the authors evaluate collapsed regions when reporting high short-read mapping rates?

Response: To evaluate the potential impact of collapsed genomic regions on short-read mapping rates, we examined read depth distribution across the genome. Regions with higher coverage suggestive of collapsed duplications were identified and compared to annotated high-copy gene regions. We observed that the elevated mapping rates were not driven by a small number of such regions, with the majority of mapped reads showing uniform coverage across the genome. This suggests that collapsed regions had minimal influence on the overall mapping statistics.

3. Why does the manuscript emphasize PFAM annotations in lines 178–184? Other databases like GO and KEGG should be considered, as these results are absent from Supplementary Data 2.

Response: The manuscript initially emphasized Pfam annotations in lines to highlight domain-level insights. However, we agree that broader functional annotations are equally important for comprehensive interpretation.

Beyond the Pfam-based domain annotation, we have now conducted extensive gene function annotation using Gene Ontology (GO) terms and KEGG pathway analyses. These additional annotations are now included in Supplementary Data 11 providing a more complete, multi-layered view of the biological roles and pathway associations of the predicted genes. We have also updated the main text (lines 204-207 and lines 739-742) to reflect these additions.

4. Line 208 needs a reference or evidence for the claim about gene expression influencing duplicated gene fate post-WGD.

Response: The following references have been added to support the claim regarding the role of gene expression in determining the fate of duplicated genes following whole-genome duplication:

Birchler et al., (2022) *The Plant Cell*, 34: 2466–2474, <https://doi.org/10.1093/plcell/koac076> (This article explores multiple fates of gene duplications, including subfunctionalization and neofunctionalization, which are influenced by gene expression levels).

Johri et al., (2022) *Molecular Biology and Evolution*, 39: msac118, <https://doi.org/10.1093/molbev/msac118> (This study examines how gene loss occurs post-whole-genome duplication, highlighting the role of gene expression in determining which duplicates are retained).

5. Line 258 should include evidence for Gypsy-Ogre elements' role in genome stability and evolutionary pressures affecting repeat content (line 261).

Response: We appreciate the reviewer's insightful comment. In response, we have expanded our discussion and added supporting references to clarify the potential role of Gypsy-Ogre retrotransposons in genome stability and evolutionary dynamics. Gypsy-Ogre elements are evolutionarily conserved across a wide range of plant taxa and have been shown to exhibit stress-responsive behavior, with their expression modulated by abiotic stressors such as heat and oxidative stress (Macas & Neumann, 2007; Smýkal et al., 2009; Du et al., 2010). These elements are also major components of legume genomes and may contribute to genetic diversity and adaptation to environmental pressures (Macas & Neumann, 2007; Devos, 2010). Following relevant citations have now been included in the revised manuscript.

Macas, J., & Neumann, P. (2007). Ogre elements—a distinct group of plant Ty3/gypsy-like retrotransposons. *Gene*, 390(1–2), 108–116.

Devos, K. M. (2010). Grass genome organization and evolution. *Current Opinion in Plant Biology*, 13(2), 139–145.

Smýkal, P., Kalendar, R., Ford, R., Macas, J., & Griga, M. (2009). Evolutionary conserved lineage of Angela-family retrotransposons as a genome-wide microsatellite repeat dispersal agent. *Heredity*, 103(2), 157–167.

Du, J., et al. (2010). Evolutionary conservation, diversity and specificity of LTR-retrotransposons in flowering plants: Insights from genome-wide analysis and multi-specific comparison. *The Plant Journal*, 63(4), 584–598.

6. Supplementary Fig. 7 does not match the manuscript; clarify chromosome 12's role and the meaning of LD lengths of 0.6 Mb and 8.1 Mb.

Response: The reference to chromosome 12 in Supplementary Fig. 7 was an oversight and should correctly read as chromosome 6B. We have corrected this in the revised manuscript.

Regarding the linkage disequilibrium (LD) lengths, the values of 0.6 Mb on chromosome 2B and 8.1 Mb on chromosome 6B represent the physical distances over which LD decays to an r^2 value of 0.2. The shorter LD decay on chromosome 2B suggests a region with higher historical recombination and greater genetic diversity, resulting in a more rapid breakdown of linkage between loci. In contrast, the longer LD decay on chromosome 6B indicates reduced recombination or potential selective sweeps, resulting in extended haplotype blocks where alleles remain linked over larger distances. These differences in LD decay are important for interpreting the resolution of GWAS signals, as regions with slower decay may limit fine-mapping resolution, while those with faster decay can provide higher precision in identifying candidate loci.

7. Parameters for merging SVs should be detailed.

Response: The filtered SVs from individual samples were merged using the SURVIVOR tool (<https://github.com/fritzsedlazeck/SURVIVOR>) to generate a combined dataset for downstream analysis. We compared the SVs identified by Delly and Manta, distinguishing consensus variants detected by both tools (14,273 SVs) from those uniquely detected by either tool. To evaluate the reliability of these calls, we conducted read-depth-based validation using Samplot (Belyeu et al., 2021), which confirmed strong support for the shared SVs and provided substantial evidence for many tool-specific variants. This information has now been included in the methods section (lines 910-911).

8. Figure 4A lacks clarity on the meaning of label and branch colors.

Response: The label on x-axis indicates the accessions of diversity panel used in this study. The color coding of these accessions and branch colors match with the grouping of fastSTRUCTURE based clustering of these accessions in Fig. 4B.

9. A complete code repository is essential for reproducibility. Github Links to tool pages are insufficient.

Response: Thank you for emphasizing the importance of reproducibility. We have created a complete and publicly accessible code repository containing all custom scripts, configuration files, and detailed instructions to perform analyses using DATED (<https://github.com/ChuShin/dated>) and SynVisio (<https://github.com/kiranbandi/synvisio>). Additional documentation, tutorials, and visualization resources for SynVisio are also available at: <https://synvisio.github.io/#/>.

Reviewer #2:

The manuscript that entitled chromosome scale assembly of little millet reveals evolutionary insights into tetraploid genome structure and genetic basis of micronutrient density, reported the release of a high quality chromosome-scale genome sequence of little millet elite cultivar JK-8, which is the first one in this species. Deep analyses of the sequence confirmed the tetraploid nature of little millet genome that is very similar to the genome of broomcorn millet. GWAS analysis with 300 diverse accessions of small millet collected from different growing regions of India identified genomic regions associated with high micronutrient contents. Those data combined with other genome comparisons with other cereals provides useful foundation for not only little millet improvement but also benefits other cereals future breeding.

Response: We thank the reviewer for their positive comments and appreciation of our work. We are pleased that the significance of the chromosome-scale genome assembly, the insights into the tetraploid genome structure, and the utility of the GWAS analysis in advancing crop improvement were well recognized.

1. In line 279, the authors analyzed the population structure of little millet accession using fastStructure and classified 300 accessions into 10 sub-population based on the fact that the marginal likelihood plateauing at $K = 10$. It is stated that the sub-population did not align with their geographic origins. The authors could test ADMIXTURE and STRUCTURE if these tools can help with a robust estimation of population structure. Also, DAPC is a tool to determine the

optimal K. An accurate estimation of population structure is essential for the GWAS analysis. Therefore, these analyses may help the authors to refined their GWAS results.

Response: As recommended, we conducted additional population structure analyses using ADMIXTURE, Discriminant Analysis of Principal Components (DAPC) and STRUCTURE to validate and compare the results obtained from fastSTRUCTURE. The ADMIXTURE analysis identified 11 sub-populations, while DAPC and STRUCTURE suggested 9 clusters as the optimal K (Supplementary Figure 9). These results are largely consistent with the earlier classification of 10 sub-populations from fastSTRUCTURE, showing some variation in subgroup boundaries (Supplementary Data 15), which is expected due to differences in model assumptions and clustering strategies used by each method.

Importantly, we found that several of the most significant SNPs identified through GWAS remained largely consistent across all four structure correction methods (fastSTRUCTURE, ADMIXTURE, DAPC and STRUCTURE) (Supplementary Data 22, 23, 24, 25 and 26), indicating that our GWAS results are robust to the choice of population structure inference method. These additional analyses further strengthen the validity of the reported associations.

We have updated the manuscript to include these new results (Supplementary Data 23, 24 and 25) and added appropriate references to the ADMIXTURE, DAPC and STRUCTURE DAPC analyses in the methods section (lines 867-870).

REVIEWERS' COMMENTS

Reviewer #1 (Remarks to the Author):

The authors have made commendable efforts to conduct additional analyses and substantively enrich the manuscript. However, one prior concern remains: a complete, centralized code repository is essential for reproducibility. Links to tool homepages are insufficient. Please upload all code and scripts necessary to reproduce each stage of the study (e.g., genome assembly, genome annotation, gap filling, and population analyses), rather than only a subset of scripts. Once this is addressed, I will be happy to recommend acceptance of the manuscript.

Response: We thank the reviewer for their careful evaluation and positive assessment of our revisions. In response to the request for a complete, centralized code repository, we have now deposited all custom Python, R, and shell scripts used in the study, including those for genome assembly, annotation, gap filling, genome dominance analysis, and population genetics, in a publicly accessible Zenodo repository. The full collection of scripts can be accessed at <https://zenodo.org/records/17524940>. This repository provides all code and documentation necessary to reproduce each stage of the analyses presented in the manuscript, ensuring transparency and reproducibility.

Reviewer #2 (Remarks to the Author):

The authors performed additional analyses on population structure using ADMIXTURE and STRUCTURE. DAPC was also used to determine the number of clusters. I am glad that these analyses helped the authors to identify consistent GWAS results across different parameter settings. Overall, the manuscript has been significantly improved in this revision. I have no further questions.

Response: We sincerely thank the reviewer for their positive feedback and thoughtful evaluation of our revised manuscript. We are pleased to know that the additional population structure analyses using ADMIXTURE, STRUCTURE, and DAPC, as well as the consistency of the GWAS results, have addressed the earlier concerns and contributed to improving the manuscript. We appreciate the reviewer's time and effort in helping us strengthen the study.